# COMPLETE AND CONTINUOUS REPRESENTATIONS OF EUCLIDEAN GRAPHS

## ABSTRACT

Euclidean graphs have unordered vertices and non-intersecting straight-line edges in any Euclidean space. Our main application is for molecular graphs with vertices at atomic centers and edges representing inter-atomic bonds. Euclidean graphs are considered equivalent if they are related by isometry (any distance-preserving transformation). This paper introduces the strongest descriptors that are provably (1) invariant under any isometry, (2) complete and sufficient to reconstruct any Euclidean graph up to isometry, (3) Lipschitz continuous so that perturbations of all vertices within their $\varepsilon$-neighborhoods change the complete invariant up to a constant multiple of $\varepsilon$ in a suitable metric, and (4) computable (both invariant and metric) in polynomial time in the number of vertices for a fixed dimension. These strongest invariants transparently explained a continuous structure-property landscape for molecular graphs from the QM9 database of 130K+ molecules.

## 1 MOTIVATIONS FOR COMPLETE REPRESENTATIONS OF EUCLIDEAN GRAPHS

In real life, many rigid structures from buildings to molecules are naturally represented by geometric graphs in Euclidean space $\mathbb{R}^n$. More precisely, a *Euclidean graph* or a geometric graph $G \subset \mathbb{R}^n$ is a finite set of $m$ vertices located at distinct points of $\mathbb{R}^n$ and connected by non-intersecting straight-line edges. The vertices of $G$ are unordered (unlabeled without any indices such as $1, \ldots, m$), hence forgetting all edges of $G$ gives us the vertex cloud $V(G)$ of $m$ unordered points, see Fig. 1.

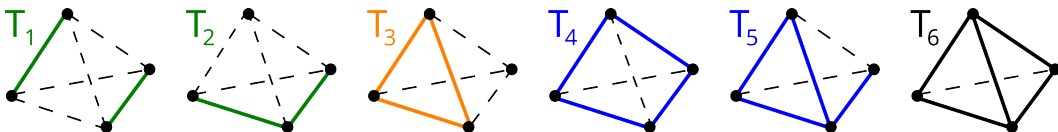

Figure 1: The Euclidean graphs $T_i \subset \mathbb{R}^3$, $i = 1, \ldots, 6$, with solid straight edges are all different but have the same four vertices of a regular tetrahedron with all pairwise distances 1, see Example 3.2.

Any Euclidean graph can be considered a geometric embedding of an abstract combinatorial graph into $\mathbb{R}^n$ so that all edges map to straight-line edges. We use the term Euclidean graph only for a straight-line graph in $\mathbb{R}^n$ because graphs can be embedded into metric spaces with other geometries such as spherical or grid-like spaces where straight lines consist of horizontal or vertical segments.

A Euclidean graph can be disconnected and can have vertices $v$ of any *degree* that is the number of edges whose endpoint is $v$. Loops and multiple edges (with the same endpoints) do not appear in a Euclidean graph only because all edges are straight and can meet only at their endpoints. However, cycles (closed paths) consisting of at least three edges are allowed. The graph of a piecewise linear function $f : [0, 1] \to \mathbb{R}$ can be considered as a Euclidean graph $G(f) = \{(x, f(x)) \mid x \in [0, 1]\} \subset \mathbb{R}^2$ for $n = 2$. Since any continuous function $f : [0, 1] \to \mathbb{R}$ can be approximated by piecewise linear ones, any road network with curved routes can be approximated by a Euclidean graph in $\mathbb{R}^2$.

The most practical cases are the low dimensions $n = 2, 3, 4$, while the number $m$ of vertices can be much larger. Our main application is for molecular graphs in $\mathbb{R}^3$, where vertices are centers of atoms and edges are inter-atomic bonds that keep atoms together in a stable enough molecule.

Any data objects including molecular graphs are always considered up to some equivalence that satisfies the axioms: any $G \sim G$, if $G \sim G'$ then $G' \sim G$, if $G \sim G'$ and $G' \sim G''$ then $G \sim G''$.

In computer science, the traditional equivalence of graphs is an *isomorphism* that bijectively maps all vertices and edges. The graph isomorphism problem still has no polynomial-time solution, see Grohe & Schweitzer (2020). In chemistry, the simplest equivalence of molecular graphs is defined by composition of chemical elements. Fig. 2 shows classical stereoisomers whose different conformations in space have equivalent compositions and isomorphic graphs but substantially differ by properties. These equivalences are weaker than *rigid motion* (a composition of translations and rotations), or *isometry*, which is any distance-preserving transformation and also includes reflections.

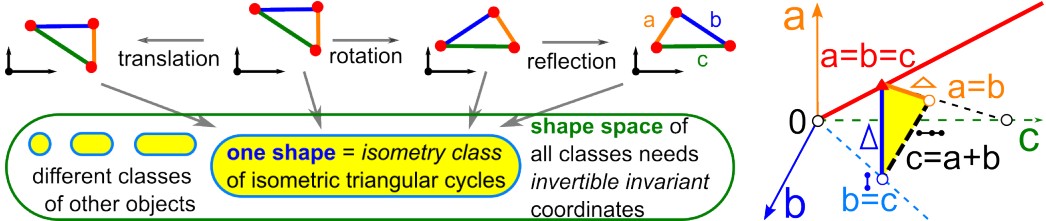

Figure 2: The graphs of stereoisomers are isomorphic even with atomic types but are not isometric.

Moving a molecule as a rigid body keeps all its chemical properties under the same ambient conditions. Even if a molecule (or another real object) is flexible, it makes sense to distinguish its different rigid shapes (3-dimensional conformations) because they can have different properties. Hence rigid motion is the strongest and most practical equivalence between real objects including molecules.

The classes of all Euclidean graphs on $m$ unordered vertices under rigid motion in $\mathbb{R}^n$ form the *Rigid Graph Space* $\mathrm{RGS}(\mathbb{R}^n; m)$. If we replace rigid motion with isometry, which can reverse the orientation of $\mathbb{R}^n$, the resulting *Isometry Graph Space* $\mathrm{IGS}(\mathbb{R}^n; m)$ is obtained from $\mathrm{RGS}(\mathbb{R}^n; m)$ by identifying mirror images of (rigid classes of) graphs in $\mathbb{R}^n$. These spaces are continuously infinite because almost any perturbation of vertices of $G \subset \mathbb{R}^n$ yields a slightly different (non-rigidly equivalent) graph $G'$. The SSS theorem from school geometry says that any triangles are congruent (isometric) if and only if they have the same triple of sides (pairwise distances) $a, b, c$ up to 6 permutations (only 3 cyclic permutations for rigid motion excluding reflections), see Fig. 3 (left).

Figure 3: **Left**: all isometry classes of graphs form a continuous space of shapes. **Right**: the space $\mathrm{IGS}(\mathbb{R}^n; 3)$ of triangular cycles under Euclidean isometry $\{0 < a \leq b \leq c \leq a + b\}$ continuously parametrised by inter-point distances $a, b, c$ with isosceles and degenerate triangles on the boundary.

Hence $\mathrm{IGS}(\mathbb{R}^n; 3)$ is the triangular cone $\{0 < a \leq b \leq c \leq a + b\}$ in $\mathbb{R}^3$, where the last triangle inequality $c \leq a + b$ guarantees that three distances $a, b, c$ are realisable for some $m = 3$ points.

If we exclude reflections, mirror images are distinguished in the larger space $\mathrm{RGS}(\mathbb{R}^n; 3)$, which is obtained by gluing two copies of $\mathrm{IGS}(\mathbb{R}^n; 3)$ along their boundaries representing mirror-symmetric triangles (isosceles and degenerate). We will define complete representations for $m \geq 4$ vertices.

Many graph descriptors such as atomic coordinates can easily change by translation and cannot reliably distinguish any objects under isometry. An *equivariant* (descriptor) is a function $f$ on graphs $G$ that changes under any isometry $T$ in a controllable way to $T_f(f(G))$, where the transformation $T_f$ is expressed via $T$. For example, if $f(G)$ is any fixed linear combination of vertices of $G$, e.g. the center of mass, then $T_f$ is the same linear map $f$ acting on the single point $f(G)$. In the important special case, if $T_f$ is the identity map not changing any $f(G)$, then $f$ is called an *invariant*. The sorted list of $\frac{m(m-1)}{2}$ pairwise distances between all $m$ unordered vertices is an isometry invariant.

Rigid classes of Euclidean graphs can be distinguished only by *invariants* that are descriptors preserved under any rigid motion. Indeed, only the definition of an invariant ($I(G) = I(G')$ for any $G \sim G'$) guarantees that if $I(G) \neq I(G')$ then $G \not\sim G'$ are not equivalent. Hence invariants are much rarer and more practically useful than equivariants. A complete classification requires the strongest (*complete* or injective) invariants that uniquely identify any Euclidean graph up to isometry. Such a complete invariant is similar to a DNA code that identifies any human in practice.

Figure 4: Non-invariant descriptors (e.g. atomic coordinates) cannot be used for comparison under isometry. Any rigid motion acting on an object changes its equivariant descriptors (e.g. the center of mass) by the same linear operation. Only invariants preserved under any equivalence can classify objects. The most useful invariants satisfy the extra practically important conditions in Problem 1.1.

Since all real measurements are noisy and molecular simulations also output only approximations of atomic coordinates, the new but practically important requirement for classifying Euclidean graphs is *continuity* under perturbations. All the conditions above are formalized in Problem 1.1 below.

**Problem 1.1.** *Find an invariant* $I$ : {*all Euclidean graphs on* $m$ *unordered vertices in* $\mathbb{R}^n$} $\rightarrow$ *a simpler space, where invariant values are easier to compare, such that the conditions below hold:*

*(a) completeness : graphs* $G, G'$ *are related by rigid motion in* $\mathbb{R}^n$ *if and only if* $I(G) = I(G')$;

*(b)* Lipschitz **continuity** : *for a constant* $\lambda$, *if any vertex of* $G$ *is perturbed within its* $\varepsilon$-*neighborhood, then* $I(G)$ *changes by at most* $\lambda\varepsilon$ *for any* $\varepsilon > 0$ *and a metric* $d$ *satisfying the metric axioms below:*

*1)* $d(I(G), I(G')) = 0$ *if and only if the graphs* $G \cong G'$ *are related by rigid motion in* $\mathbb{R}^n$,

*2)* symmetry : $d(I(G), I(G')) = d(I(G'), I(G))$ *for any Euclidean graphs* $G, G' \subset \mathbb{R}^n$,

*3)* triangle inequality : $d(I(G), I(G')) + d(I(G'), I(G'')) \geq d(I(G), I(G''))$ *for any* $G, G', G''$;

*(c) computability : $I$ and $d$ can be computed in polynomial time in $m$ for a fixed dimension $n$;*

*(d) reconstructability : any graph* $G \subset \mathbb{R}^n$ *can be reconstructed (up to rigid motion) from* $I(G)$.

Condition 1.1(a) means that $I$ is a descriptor with (1) *no false negatives*, which are pairs of $G \cong G'$ with $I(G) \neq I(G')$, and (2) *no false positives*, which are pairs of $G \ncong G'$ with $I(G) = I(G')$. The axioms in 1.1(b) are the foundations of metric geometry Melter & Tomescu (1984) and accepted in physical chemistry Weinhold (1975). If the first axiom fails, even the zero distance $d \equiv 0$ satisfies two other axioms. The first axiom implies the completeness of $I$ in 1.1(a) but the continuity is much stronger. Indeed, for any complete invariant $I$, one can define the discrete metric $d(I(G), I(G')) = 1$ for $G \ncong G'$, which unhelpfully treats all different graphs (even near-duplicates) as equally distant.

The Lipschitz continuity in 1.1(b) is much stronger than the classical $\varepsilon - \delta$ continuity because the Lipschitz constant $\lambda$ should be independent of a graph $G$ and $\varepsilon$. Computability 1.1(c) prevents brute-force attempts, for example, defining $I(G)$ as the infinite set of images of $G$ under all possible rigid motions or minimizing a distance $d$ between infinitely many alignments of given graphs.

Final condition 1.1(d) requires the invariant $I$ to be not only complete and continuous but also easy enough to allow an explicit reconstruction of $G \subset \mathbb{R}^n$ in polynomial time in the number $m$ of vertices for a fixed dimension $n$. A human cannot be reconstructed from their DNA code yet.

The key contribution of this paper is a mathematically justified solution to Problem 1.1 for rigid motion and related equivalences including isometry and their compositions with uniform scaling. Section 2 will review the closely related past work. Section 3 will define the Simplexwise Distance Distribution (SDD) invariant for graphs in any metric space. Section 4 will specialize SDD for $\mathbb{R}^n$ to build a final invariant satisfying all the conditions in Problem 1.1 for any Euclidean graphs.

The supplementary materials include proofs of all theorems and C++/Python code for the invariants.

## 2 PAST WORK ON GEOMETRIC CLASSIFICATIONS OF CLOUDS AND GRAPHS

Problem 1.1 is a key step in understanding the *structure-property* relationships for many real-life objects from galaxy formations to molecular graphs. Indeed, the concept of a *structure* requires a definition of an equivalence relation before such structures can be called equivalent or different.

Problem 1.1 is much simpler for a cloud $C$ of $m$ unordered points when a graph has only isolated vertices and no edges. For a given $m \times m$ distance matrix of $C$, the classical multidimensional scaling Schoenberg (1935) finds an embedding $A \subset \mathbb{R}^k$ (if it exists) preserving all distances of $M$ for a dimension $k \leq m$. This embedding $C \subset \mathbb{R}^k$ uses eigenvectors whose ambiguity up to signs gives an exponential comparison time that can be close to $O(2^m)$, not polynomial in $m$ as in 1.1(c).

**The case of ordered points** is easy because the matrix of distances $d_{ij}$ between indexed points $p_i, p_j$ allows us to reconstruct $C$ by using the known distances to the previously constructed points (Grinberg & Olver, 2019, Theorem 9). For clouds of $m$ ordered points, the difference between $m \times m$ matrices of distances (or Gram matrices of $p_i \cdot p_j$) can be converted into a continuous metric by taking a matrix norm. If given points are unordered, comparing $m \times m$ matrices needs an exponential number $m!$ of permutations, which is impractical, also for graphs in Problem 1.1.

The related problems of matching and finding distances between fixed Euclidean graphs (but not for their isometry classes) were studied in Nikolentzos et al. (2017); Majhi & Wenk (2022); Buchin et al. (2023). Equivariant descriptors of graphs Gao et al. (2020); Qi & Luo (2020); Tu et al. (2022); Batzner et al. (2022), e.g. the center of mass, may not be isometry invariants. Hence all conditions of Problem 1.1 are much stronger and were not proved for any past invariants of Euclidean graphs.

**Geometric Deep Learning** in Bronstein et al. (2021) pioneered an axiomatic approach to geometric classifications and went beyond Euclidean space $\mathbb{R}^n$ in Bronstein et al. (2017), so section 3 considers Problem 1.1 in any metric space. The axioms in 1.1(b) are important not only because they are basic requirements for any proofs in metric geometry, see Dorst et al. (2010), but also because if the triangle axiom fails with any additive error or missed as in (Koutra et al., 2013, section 2.4), classical $k$-means and DBSCAN clustering are open to adversarial attacks in Rass et al. (2022).

The words "Euclidean graphs" and "graph isomorphisms" are sometimes used Hordan et al. (2023) for clouds of unordered points without edges: "we discuss geometric graph isomorphism tests, namely, tests for checking whether two given point clouds $X, Y \in \mathbb{R}^{3m}$ are related by a permutation, rotation and translation" (Hordan et al., 2023, section 2). The notation $X \in \mathbb{R}^{3m}$ implies that all coordinates (hence, points) of $X$ are ordered, while a cloud of $m$ unordered points lives in a quotient of $\mathbb{R}^{3m}$ by $m!$ permutations. All known descriptors (sometimes called fingerprints) of molecular graphs Duvenaud et al. (2015); Choo et al. (2023) have no proofs of conditions 1.1(a,b,c,d).

**Energy potentials** are often claimed to be complete representations of atomic environments, though this completeness holds only for infinite-size representations Pozdnyakov et al. (2020). The review in (Kulik et al., 2022, section 1.5.2) highlights that "it is unclear which representation types produce the most accurate and transferable deep learning models". The completeness in Rose et al. (2023) for the Weisfeiler-Lehman test still needs proof of Lipschitz continuity. This paper extends the recent solution of simplified Problem 1.1 for Euclidean clouds in Widdowson & Kurlin (2023) to graphs.

**Structure-property relationship** (SPR) hypothesis says that structure determines all properties Trolier-McKinstry & Newnham (2018). The solution to Problem 1.1 finally settles the concept of a *geometric structure* (for molecular graphs) in the sense that properties can now be studied in terms of the complete invariants instead of incomplete descriptors. Since all real structures differ at least slightly because of noise, Problem 2.1 below is a more practical continuous version of SPR.

**Problem 2.1.** *For any real-valued property of a molecule, find the upper bounds of $\varepsilon$ and $\lambda$ such that perturbing any graph $G$ up to $\varepsilon$ in a metric $d$ from Problem 1.1 changes the property up to $\lambda\varepsilon$.*

In the past, many molecular properties were predicted by optimizing millions of parameters through high-cost training on large datasets with pre-computed properties, see Pinheiro et al. (2020). Since atomic coordinates have continuous real values, the space $\mathrm{RGS}(\mathbb{R}^3; m)$ of molecular shapes is infinitely continuous even for a fixed number $m$ of atoms. Since any finite dataset is a discrete sample of the continuous space $\mathrm{RGS}(\mathbb{R}^3; m)$, all sample-based predictions are hard to generalize for the full $\mathrm{RGS}(\mathbb{R}^3; m)$. Hence any new prediction algorithm may incrementally improve the past accuracy but even error 0 on a fixed dataset will not guarantee approximately corrects output on new data.

The proposed solutions to Problems 1.1 and 2.1 overcome this data limitation challenges by exploring $\mathrm{RGS}(\mathbb{R}^3; m)$ as a mountainous landscape. A metric $d$ from Problem 1.1 allowed us to find the previously unknown (hundreds of) duplicates and analyze the nearest neighbors of any molecule within the QM9 dataset of 130K+ molecules in Ramakrishnan et al. (2014). These nearest neighbors provided the first-ever bounds for Problem 2.1 and enabled the identification of deep energy minima (the most stable molecules) surrounded by energy barriers, see the experiments in section 5.

## 3 CONTINUOUS ISOMETRY INVARIANTS OF GRAPHS IN ANY METRIC SPACE

Inspired by Rodney Brooks' message "geometry is your friend" of his plenary lecture at CVPR 2023, this geometric section defines isometry invariants of a graph $G$ in any metric space $X$ with a distance metric $d$, see Fig. 5. We assume that $G$ has no loops (edges with a single endpoint) and no multiple edges. Any vertex $p$ and unordered pair $[p, q]$ of vertices of $G$ can have an *attribute* $a(p)$ and a *weight* $w[p, q]$, which should be respected by any isometry that maps one graph to another.

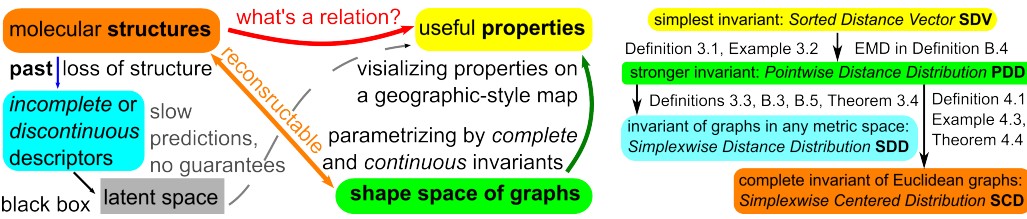

Figure 5: **Left**: structure-property relations. **Right**: connections between invariants in sections 3, 4.

Any metrics on vertex attributes and weights can be incorporated into metrics on the invariants below. To make the key concepts clearer, the main paper introduces all invariants for unordered vertices without attributes and only for sign weights: $w[p, q] = +1$ if the vertices $p, q$ are connected by an edge of $G$, otherwise $w[p, q] = -1$. Let $d_G(p, q)$ denote any *G-based distance* between $p, q \in V(G)$. We can take the *signed* distance $w[p, q]d(p, q)$ or the *shortest path* distance in $G$.

**Definition 3.1** (invariants SDV and PDD). *Let $G \subset X$ be any graph with $m$ vertices $p_1 \dots, p_m$,*

*(a) The* Sorted Distance Vector $\text{SDV}(G)$ *is the list of $\frac{m(m-1)}{2}$ G-based distances $d_G(p_i, p_j)$ written in increasing order, hence starting from 'negative' values for the signed distance $w[p, q]d(p, q)$.*

*(b) Let $\text{DD}(G)$ be the $m \times (m - 1)$-matrix where the $i$-th rows consists of G-based distances $d_G(p_i, p_j)$, $j \in \{1, \dots, m\} - \{i\}$, written in increasing order. If $k > 1$ rows of $\text{DD}(G)$ are identical to each other, we collapse them into a single row with the weight $k/m$. Write the weight of any row in the extra first column and call the resulting matrix the* Pointwise Distance Distribution $\text{PDD}(G)$.

$\text{PDD}(G)$ includes every G-based distance twice, once as $d_G(p, q)$ in the row of a vertex $p$, and as $d_G(q, p)$ in the row of a vertex $q$. Hence $\text{SDV}(G)$ can be obtained from $\text{PDD}(G)$ by (1) combining all distances into one vector, (2) sorting them in increasing order, and (3) keeping only one copy of every two repeated distances. Example 3.2 shows that $\text{PDD}(G)$ is stronger than $\text{SDV}(G)$.

**Example 3.2** (SDV and PDD with signed distances for tetrahedral graphs in Fig. 1). *(a) The vertices of the six tetrahedral graphs $T_i \subset \mathbb{R}^3$, $i = 1, \dots, 6$ in Fig. 1 have all pairwise distances 1. All these graphs have the same set of vertices $V(T_i)$ independent of $i$, hence cannot be distinguished by isometry invariants of $V(T_i)$. The first graph $T_1$ has two edges contributing $+1$ and four non-edges (dashed lines) contributing $-1$ to the Sorted Distance Vector $\text{SDV}(T_1) = (-1, -1, -1, -1, +1, +1)$. The graph $T_2$ also has two edges, so $\text{SDV}(T_2) = \text{SDV}(T_1)$ doesn't distinguish $T_1 \not\cong T_2$ up to isometry. Finally,*

$$\text{SDV}(T_3) = (-1, -1, -1, +1, +1, +1)$$
$$\text{SDV}(T_4) = (-1, -1, +1, +1, +1, +1) = \text{SDV}(T_5).$$
$$\text{SDV}(T_6) = (-1, +1, +1, +1, +1, +1)$$

*(b) In $T_1$, every vertex has exactly one edge and two non-edges (dashed lines), hence its signed distances are $-1, -1, +1$. The matrix $\text{PDD}(T_1) = (100\% \mid -1, -1, +1)$ consists of a single row, where the weight $100\%$ indicates that all vertices of $T_1$ have the same row in PDD. The graph $T_2$ has one vertex (25%) with no edges, two vertices (50%) with one edges, and one vertex (25%) with two edges, so* $\text{PDD}(T_2) = \begin{pmatrix} 25\% & -1 & -1 & -1 \\ 50\% & -1 & -1 & +1 \\ 25\% & -1 & +1 & +1 \end{pmatrix} \neq \text{PDD}(T_1)$, *so* PDD *distinguishes the non-isometric graphs $T_1 \not\cong T_2$ with $\text{SDV}(T_1) = \text{SDV}(T_2)$. The graph $T_3$ has one vertex (25%) with no edges and three vertices (75%) with two edges, so* $\text{PDD}(T_3) = \begin{pmatrix} 25\% & -1 & -1 & -1 \\ 75\% & -1 & +1 & +1 \end{pmatrix}$.

*Then* $\mathrm{PDD}(T_4) = (100\% \mid -1, +1, +1) \neq \mathrm{PDD}(T_5) = \begin{pmatrix} 25\% & -1 & -1 & +1 \\ 50\% & -1 & +1 & +1 \\ 25\% & +1 & +1 & +1 \end{pmatrix}$, *so* PDD *dis-*

*tinguishes* $T_4 \not\cong T_5$ *with* $\mathrm{SDV}(T_4) = \mathrm{SDV}(T_5)$. *Finally,* $\mathrm{PDD}(T_6) = \begin{pmatrix} 50\% & -1 & +1 & +1 \\ 50\% & +1 & +1 & +1 \end{pmatrix}$.

For a graph $G$ with $m$ vertices, the full matrix $\mathrm{PDD}(G)$ consists of $m-1$ columns but can consider the reduced version $\mathrm{PDD}(G; k)$ including only the first $k$ columns for any $1 \leq k < m-1$. Though the matrices $\mathrm{PDD}(G)$ can have different sizes, any of them can be continuously compared as discrete probability distributions, for example, by Earth Mover's Distance EMD, which we remind in Definition B.5, see Rubner et al. (2000). The rows of $\mathrm{PDD}(G)$ are unordered, so EMD is invariant under their permutation, though we write them in the lexicographic order only for convenience.

The *lexicographic* order $u < v$ on vectors $u = (u_1, \ldots, u_h)$ and $v = (v_1, \ldots, v_h)$ in $\mathbb{R}^h$ means that if the first $i$ (possibly, $i = 0$) coordinates of $u, v$ coincide then $u_{i+1} < v_{i+1}$. For example, $(1, 2) < (2, 1) < (2, 2)$. Let $S_h$ denote the permutation group on $1, \ldots, h$. Definition 3.3 extends $\mathrm{PDD}(G)$ to $\mathrm{SDD}(G; h)$ for graphs $G$ in a metric space, where $\mathrm{SDD}(G; 1) = \mathrm{PDD}(G)$.

**Definition 3.3** (Simplexwise Distance Distribution $\mathrm{SDD}(G; h)$). *Let $G$ be a graph on $m$ unordered vertices in a space with a metric $d$. Let $A = (p_1, \ldots, p_h) \subset V(G)$ be an ordered subset of $1 \leq h < m$ vertices. Let $D(A)$ be the matrix whose entry $D(A)_{i,j-1}$ is the distance $d_G(p_i, p_j)$ for $1 \leq i < j \leq h$, all other entries are filled by zeros. Any permutation $\xi \in S_h$ acts on $D(A)$ by mapping $D(A)_{ij}$ to $D(A)_{kl}$, where $k \leq l$ is the pair of indices $\xi(i), \xi(j) - 1$ written in increasing order. For any $q \in V(G) - A$, write the $G$-based distances from $q$ to $p_1, \ldots, p_h$ as a column.*

*The $h \times (m-h)$-matrix $R(G; A)$ is formed by these $m - h$ lexicographically ordered columns. The action of $\xi$ on $R(G; A)$ maps any $i$-th row to the $\xi(i)$-th row, then all columns can be written in the lexicographic order. The* Relative Distance Distribution $\mathrm{RDD}(G; A)$ *is the equivalence class of the pair $[D(A), R(G; A)]$ of matrices up to actions of $\xi \in S_h$. The* Simplexwise Distance Distribution $\mathrm{SDD}(G; h)$ *is the unordered set of $\mathrm{RDD}(C; A)$ for all unordered $h$-vertex subsets $A \subset V(G)$.*

For a 1-point subset $A = \{p_1\}$ with $h = 1$, the matrix $D(A)$ is empty and $R(G; A)$ is one row of distances (in increasing order) from $p_1$ to all other vertices $q \in V(G)$. For a 2-point subset $A = (p_1, p_2)$ with $h = 2$, the matrix $D(A)$ is the single number $d_G(p_1, p_2)$ and $R(G; A)$ consists of two rows of distances from $p_1, p_2$ to all other points $q \in V(G)$. Unordered collections of the same size can be compared by the bottleneck match, which we adapt to the Simplexwise Bottleneck Metric (SBM) on SDDs, see Definition B.3. If we collapse any $l > 1$ identical RDDs into a single RDD with the *weight* $l/\binom{m}{h}$, SDD can be considered as a weighted probability distribution of RDDs. Any finite distributions of different sizes can be compared by the EMD in Definition B.5.

Theorems 3.4 and 4.4 will extend the $O(m^{1.5} \log^n m)$ algorithms for fixed clouds of $m$ unordered points in (Efrat et al., 2001, Theorem 6.5) to the harder case of isometry classes but keep polynomial time in $m$ for a fixed dimension $n$. All complexities are for a random-access machine (RAM) model.

**Theorem 3.4** (computability and continuity of SDDs). *(a) For $h \geq 1$ and a graph $G$ on $m$ unordered vertices, $\mathrm{SDD}(C; h)$ is an isometry invariant and can be computed in time $O(m^{h+1}/(h-1)!)$.*

*(b) For any graphs $G, G'$ on $m$ vertices and $h \geq 1$, the metric SBM from Definition B.3 satisfies all metric axioms on SDDs and can be computed in time $O\left((m^{2.5h} + m^{2h+1.5} \log^h m)/h!\right)$.*

*(c) For any graphs $G, G'$ and $h \geq 1$, let $\mathrm{SDD}(G; h)$ and $\mathrm{SDD}(G'; h)$ have a maximum size $l \leq \binom{m}{h} = O(m^h/h!)$ after collapsing identical RDDs. Then EMD from Definition B.5 satisfies all metric axioms on SDDs and can be computed in time $O(h!(h^2 + m^{1.5} \log^h m)l^2 + l^3 \log l)$.*

*(d) In a metric space, perturbing any vertex of a graph $G$ within its $\varepsilon$-neighborhood changes $\mathrm{SDD}(G; h)$ up to $2\varepsilon$ in the metrics SBM and EMD from Definitions B.3 and B.5, respectively.*

Theorem 3.4(d) substantially generalizes the fact that perturbing two points in their $\varepsilon$-neighborhoods changes the distance between these points by at most $2\varepsilon$. We conjecture that $\mathrm{SDD}(G; h)$ is a complete isometry invariant of all Euclidean graphs $G \subset \mathbb{R}^n$ for some $h \geq n - 1$. If $G$ is a complete graph on a cloud $C$, (Kurlin, 2023, section 4) shows that $\mathrm{SDD}(C; 2)$ distinguished all infinitely many known pairs in (Pozdnyakov et al., 2020, Fig. S4) of non-isometric $m$-point clouds $C, C' \subset \mathbb{R}^3$ that have the same $\mathrm{PDD}(C) = \mathrm{SDD}(C; 1)$. Section 4 extends SDD to a complete invariant SCD in $\mathbb{R}^n$.

# 4  COMPLETE INVARIANTS OF EUCLIDEAN GRAPHS UNDER RIGID MOTION

The Simplexwise Distance Distribution SDD from section 3 is based only on distances, hence cannot distinguish mirror images of graphs in $\mathbb{R}^n$. This section adapts SDD to a complete and continuous invariant of Euclidean graphs under rigid motion in $\mathbb{R}^n$ by including signs that can detect orientation.

First, for any Euclidean graph $G \subset \mathbb{R}^n$, we translate the *center of mass* $\frac{1}{m} \sum_{p \in V(G)} p$ of the vertex set $V(G)$ to the origin $0 \in \mathbb{R}^n$. Below we develop metrics on complete invariants of $G \subset \mathbb{R}^n$ up to rotations around the origin $0 \in \mathbb{R}^n$. In some applications, molecular graphs can include a specific heavy atom, which can be more convenient to choose for the origin instead of the center of mass.

For a molecular graph $G$ in $\mathbb{R}^n$, the simplest choice of a $G$-based distance is $w(p, q)|p - q|$, where $w(p, q) = 1$ for bonded atoms and $w(p, q) = -1$ for non-bonded atoms $p, q$. We use inter-atomic bonds in a smarter way to use the shortest path length $d_G(p, q)$ along the centered graph $G \cup \{0\}$, where the center $0$ of mass is connected to all atoms, and every edge has the Euclidean distance.

For any ordered sequence $A = \{p_1, \dots, p_{n-1}\} \subset V(G)$, the distance matrix $D(A \cup \{0\})$ in Definition 3.3 has size $(n - 1) \times (n - 1)$ and its last column can be chosen to include the distances from the origin $0 \in \mathbb{R}^n$ to the $n - 1$ vertices of $A$. So $0$ is treated as an extra isolated vertex of $G$ even if one $p_i = 0$. Any $n$ vectors $v_1, \dots, v_n \in \mathbb{R}^n$ can be written as columns in the $n \times n$ matrix whose determinant has a *sign* $\pm 1$ or $0$ if $v_1, \dots, v_n$ are linearly dependent. Any permutation $\xi \in S_n$ on indices $1, \dots, n$ is a composition of some $t$ transpositions $i \leftrightarrow j$ and has $\text{sign}(\xi) = (-1)^t$.

**Definition 4.1** (Simplexwise Centered Distribution SCD). *Let $G \subset \mathbb{R}^n$ be any Euclidean graph on $m$ unordered vertices and center of mass $0 \in \mathbb{R}^n$. For any ordered subset $A$ of vertices $p_1, \dots, p_{n-1} \in V(G)$ and any $q \in V(G) - A$, the matrix $R(C; A \cup \{0\})$ has the column of the distances $d_G(q, p_1), \dots, d_G(q, p_{n-1}), d_G(q, 0)$, where $d$ is the Euclidean distance. At the bottom of this column, insert the sign of the determinant of the $n \times n$ matrix whose columns are $q - p_1, \dots, q - p_{n-1}, q$. The resulting $(n+1) \times (m-n+1)$-matrix is the* oriented relative distance matrix $M(G; A \cup \{0\})$. Any permutation $\xi \in S_{n-1}$ of $n-1$ points of $A$ acts on $D(A)$, permutes the first $n-1$ rows of $M(G; A \cup \{0\})$ and multiplies every sign in the $(n+1)$-st row by $\text{sign}(\xi)$. The Oriented Centered Distribution $\text{OCD}(G; A)$ is the equivalence class of pairs $[D(A \cup \{0\}), M(G; A \cup \{0\})]$ considered up to permutations $\xi \in S_{n-1}$ of points of $A$. The Simplexwise Centered Distribution $\text{SCD}(G)$ is the unordered set of the distributions $\text{OCD}(G; A)$ for all $\binom{m}{n-1}$ unordered $(n-1)$-vertex subsets $A \subset V(G)$. The mirror image $\overline{\text{SCD}}(G)$ is obtained from $\text{SCD}(G)$ by reversing all signs.*

Definition 4.1 needs no permutations for $G \subset \mathbb{R}^2$ as $n - 1 = 1$. Columns of $M(G; A \cup \{0\})$ can be lexicographically ordered without affecting the metric in Lemma C.4. If we collapse any $l > 1$ equal OCDs into a single OCD with the *weight* $l/\binom{m}{h}$, then SCD can be considered a weighted probability distribution of several OCDs. To get a continuous metric on OCDs and then define SBM, EMD on SCDs, we will multiply each sign by a continuous *strength* function below.

**Definition 4.2** (strength $\sigma(A)$ of a simplex). *For a set $A$ of $n+1$ points $q = p_0, p_1, \dots, p_n$ in $\mathbb{R}^n$, let $p(A) = \frac{1}{2} \sum_{i \neq j} |p_i - p_j|$ be half of the sum of all pairwise distances. Let $V(A)$ denote the volume the $n$-dimensional simplex on the set $A$. The strength is defined as $\sigma(A) = V^2(A)/p^{2n-1}(A)$.*

For $n = 1$ and a set $A = \{p_0, p_1\} \subset \mathbb{R}$, the volume is $V(A) = |p_0 - p_1| = 2p(A)$, so $\sigma(A) = 2|p_0 - p_1|$. For $n = 2$ and a triangle $A$ with sides $a, b, c$ in $\mathbb{R}^2$, Heron's formula gives $\sigma(A) = \frac{(p-a)(p-b)(p-c)}{p^2}$, $p = \frac{a+b+c}{2} = p(A)$ is the half-perimeter of $A$. The strength $\sigma(A)$ depends only on the distance matrix $D(A)$ from Definition 3.3 but we use the notation $\sigma(A)$ for brevity. In any $\mathbb{R}^n$, the squared volume $V^2(A)$ is expressed by the Cayley-Menger determinant in distances between $n + 1$ points of a set $A$. The strength $\sigma(A)$ vanishes when the simplex on $A$ degenerates.

**Example 4.3** (SCD of tetrahedral graphs in Fig. 1). *The vertex set $V$ of each tetrahedral graph $T_i$, $i = 1, \dots, 6$, in Fig. 1 are the vertices of a regular tetrahedron $T$ that has six edges of length 1 and circumradius $R = \frac{\sqrt{6}}{4}$. Fix the center of mass of $T$ at the origin $0$ of $\mathbb{R}^3$. Let $v_T, v_L, v_B, v_R$ denote the four vertices of $T$ at the top, left, bottom, and right corners, respectively, in Fig. 1.*

*By Definition 4.1 for $n = 3$ we consider 6 unordered subsets $A$ of two vertices $p_1, p_2 \in V$. For each $A$, the distance matrix $D(A \cup \{0\})$ on the triangle $p_1, p_2, 0$ has the same distances $d_G(p_1, 0) = -R = d_G(p_1, 0)$, because $0$ is considered an extra added vertex disjoint with $T_i$. The signed distance $d_G(p_1, p_2)$ equals $+1$ if $T_i$ includes the edge $[p_1, p_2]$, otherwise $d_G(p_1, p_2) = -1$.*

Graph $T_1$. *Case $A = (p_1, p_2) = (v_T, v_L)$. The vertices $v_T, v_L$ are joined by an edge in $T_1$, so* $d_G(p_1, p_2) = +1$ *and* $D(A) = \begin{pmatrix} +1 & -R \\ * & -R \end{pmatrix}$, *where we show only the upper triangle above the*

*zero diagonal in the full symmetric $3 \times 3$ matrix. The matrix $M(T_1; A \cup \{0\}) = \begin{pmatrix} -1 & -1 \\ -1 & -1 \\ -R & -R \\ + & - \end{pmatrix}$*

*has four rows (one for each of $p_1, p_2, 0$ and signs of determinants) and two columns (one for each of two other vertices $v_B, v_R$). For $A = (p_1, p_2) = (v_B, v_R)$, we get the same Oriented Centered Distribution $\mathrm{OCD}(G; A)$ represented by the pair $[D(A \cup \{0\}), M(G; A \cup \{0\})]$ above due to the symmetry $v_T \leftrightarrow v_B$, $v_L \leftrightarrow v_R$ in the graph $T_1$. For the other four 2-vertex subsets including*

$A = (p_1, p_2) = (v_T, v_R)$, *we get* $D(A) = \begin{pmatrix} -1 & -R \\ * & -R \end{pmatrix}$ *and* $M(T_1; A \cup \{0\}) = \begin{pmatrix} +1 & -1 \\ -1 & +1 \\ -R & -R \\ + & - \end{pmatrix}$.

An equality $\mathrm{SCD}(G) = \mathrm{SCD}(G')$ is interpreted as a bijection between unordered sets $\mathrm{SCD}(G) \to \mathrm{SCD}(G')$ matching all OCDs, which is best checked by $\mathrm{SBM} = 0$ or $\mathrm{EMD} = 0$ in Theorem 4.4.

Let a $G$-based distance $d_G(p, q)$ have a Lipschitz constant $\lambda$ so that perturbing $p, q$ up to $\varepsilon$ changes $d_G(p, q)$ up to $\lambda \varepsilon$. If $d_G$ is the Euclidean or signed distance, then $\lambda = 2$. If $d_G$ is the shortest path distance in the centered graph $G \cup \{0\}$, then $\lambda = 4$ because any $p, q \in V(G)$ are joined via 0.

**Theorem 4.4** (completeness and continuity of SCD). *(a) The Simplexwise Centered Distribution $\mathrm{SCD}(G)$ is a complete isometry invariant of a Euclidean graph $G \subset \mathbb{R}^n$ on $m$ unordered vertices with a center of mass at the origin $0 \in \mathbb{R}^n$, and can be computed in time $O(m^n/(n-4)!)$. So any Euclidean graphs $G, G' \subset \mathbb{R}^n$ are related by rigid motion (isometry, respectively) if and only if $\mathrm{SCD}(G) = \mathrm{SCD}(G')$ ($\mathrm{SCD}(G)$ equals $\mathrm{SCD}(G')$ or its mirror image $\overline{\mathrm{SCD}}(G')$, respectively).*

*(b) For any graphs $G, G'$ on $m$ vertices in $\mathbb{R}^n$, the metric $\mathrm{SBM}(G, G')$ from Definition C.5 satisfies all metric axioms and can be computed in time $O\left((m^{2.5(n-1)} + m^{2n-0.5}\log^n m)/(n-1)!\right)$.*

*(c) Let $\mathrm{SCD}$s have a maximum size $l \leq \binom{m}{n-1}$ after collapsing equal OCDs. Then $\mathrm{EMD}$ satisfies all metric axioms on $\mathrm{SCD}$s and can be computed in time $O((n-1)!(n^2 + m^{1.5}\log^n m)l^2 + l^3 \log l)$.*

*(d) In $\mathbb{R}^n$, perturbing any vertex of a graph $G$ within its $\varepsilon$-neighborhood changes $\mathrm{SCD}(G)$ up to $\lambda \varepsilon$ in the metrics $\mathrm{SBM}$ and $\mathrm{EMD}$ from Definitions C.5 and B.5, respectively.*

## 5 EXPERIMENTS ON MOLECULES AND A DISCUSSION OF SIGNIFICANCE

Isabelle Guyon's keynote at NeurIPS 2022 highlighted that no papers would be accepted if all error bars were included. This paper is very different due to complete solutions, not incremental improvements. Almost all past work on the QM9 database of 130K+ (130,808) molecules focused on lowering the errors of property predictions by optimizing millions of parameters, see Pinheiro et al. (2020); Lim et al. (2022); Stärk et al. (2022). Even if the error becomes 0, this experimental outcome is restricted to a specific dataset. The real problem is to *find new molecules* with the best (not yet known) properties, not to *predict the (already known) properties* of the existing molecules.

This section describes three advances that were impossible to achieve by any past tools on QM9: (1) the invariants from Problem 1.1 detected the first (near-)duplicates, (2) the Lipschitz continuity of properties vs structure is confirmed for Problem 2.1, and (3) the continuous spaces $\mathrm{RGS}(\mathbb{R}^3; m)$ of $m$-atom graphs have *detectable deepest minima* of energy surrounded by high energy barriers.

QM9 has nearly a billion (873,527,974) pairs of molecules with the same number of atoms. We started all comparisons by computing the simplest $L_\infty$ metric (max abs difference of corresponding coordinates) on SDVs, then computed 8,735,279 EMD distances on PDDs of the 1% closest pairs, and 10,000 SBM distances on SCDs of the top closest pairs. Table 1 shows many (near-)duplicates whose invariants differ only by floating-point errors in fractions of $1\text{Å} \approx$ smallest bond length.

The supplementary materials include readme.txt describing the technical specifications of the machine, running times, C++/Python code, and tables of distances within the allowed limit of 100MB.

Table 1: Pairs of near-duplicate molecules in QM9 for different thresholds and distances.

| distances/thresholds | $\leq 0.1\text{Å}$ | $\leq 0.01\text{Å}$ | $\leq 10^{-3}\text{Å}$ | $\leq 10^{-4}\text{Å}$ | $\leq 10^{-5}\text{Å}$ | $\leq 10^{-6}\text{Å}$ |
|---|---|---|---|---|---|---|
| $L_\infty$ metric on SDVs | 1315 | 235 pairs | 177 pairs | 106 pairs | 53 pairs | 11 pairs |
| EMD on PDDs | 239 | 201 pairs | 123 pairs | 66 pairs | 21 pairs | 1 pair |
| SBM on SCDs | 1334 | 239 pairs | 206 pairs | 127 pairs | 67 pairs | 7, SBM $= 0$ |

The continuous metrics are compared in Fig. 8 and allowed us to analyse QM9 within the continuous space $\cup_{m=3}^{29}\text{RGS}(\mathbb{R}^3; m)$ of $m$-atom graphs. Fig. 6 and Table 2 quantitatively justify the Lipschitz continuity of molecular properties, most importantly the free energy G, which characterizes stability.

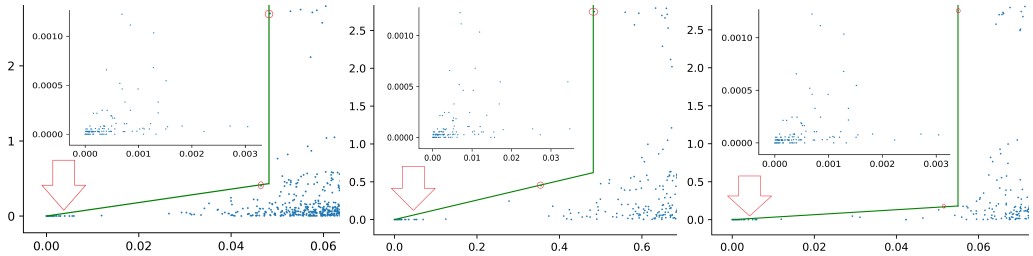

Figure 6: Continuity of the QM9 free energy vs $L_\infty$_SDV, EMD_PDD, SBM_SCD, see Table 2.

Table 2: The Lipschitz continuity of every QM9 property $P$ is confirmed in Problem 2.1 for each distance $d$ as $|P(G) - P(G')| \leq \text{SPF} \cdot d(G, G')$ for all $d \leq \text{SPD}$ (structure-property deviation).

| bounds/properties | free energy G | enthalpy H | energy U | energy U0 | energy ZPVE |
|---|---|---|---|---|---|
| SPD_$L_\infty$_SDV | 0.046 | 0.046 | 0.046 | 0.055 | 0.074 |
| SPF_$L_\infty$_SDV | 8.96 | 8.80 | 8.80 | 61.54 | 0.594 |
| SPD_EMD_PDD | 0.48 | 0.36 | 0.36 | 0.484 | 0.705 |
| SPF_EMD_PDD | 1.29 | 1.26 | 1.26 | 5.64 | 0.051 |
| SPD_SBM_SCD | 0.056 | 0.055 | 0.055 | 0.055 | 0.050 |
| SPF_SBM_SCD | 3.33 | 3.40 | 3.40 | 49.63 | 0.144 |

Fig. 7 (left) shows the energy landscape of QM9 in two well-defined coordinates $x, y$ expressed via the minimum, median, and maximum of inter-atomic distances, see other heatmaps in appendix A.

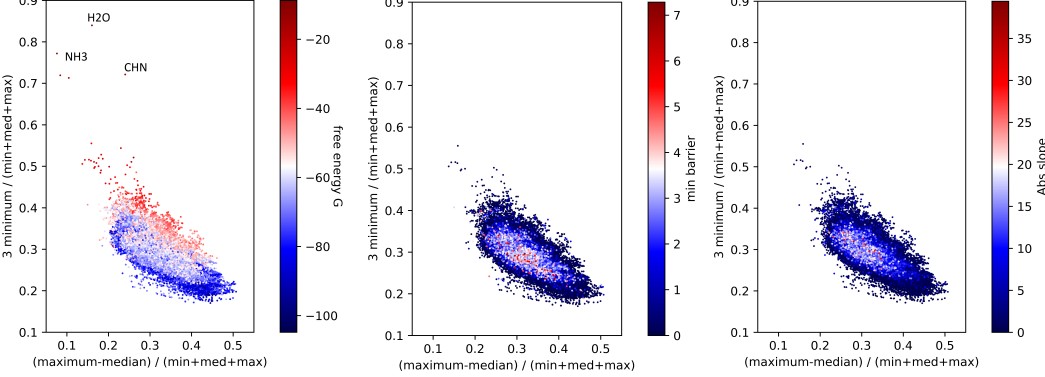

Figure 7: QM9 is projected to $x, y$ expressed via inter-atomic distances. **Left**: colored by the free energy $G$, which determines molecular stability. **Middle**: colored by the energy barrier (min difference with 5 neighbors by $L_\infty$_SDV), only 43 molecules have high barriers $\geq 5$. **Right**: colored by the min absolute slope (over 5 neighbors by $L_\infty$_SDV), another way to find deep local minima.

Fig. 7 (middle) reveals the landscape of QM9 is mostly flat and only a few molecules (shown in red) are deep energy minima. We thank all reviewers for their valuable time and helpful suggestions.

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

## A    EXTRA EXPERIMENTS ON MOLECULAR GRAPHS FROM SECTION 5

Fig. 8 shows that the new metric SBM on complete invariants SCD from Theorem 4.4 differs from the past distances on the weaker invariants SDV and PDD, which are still useful due to their speed.

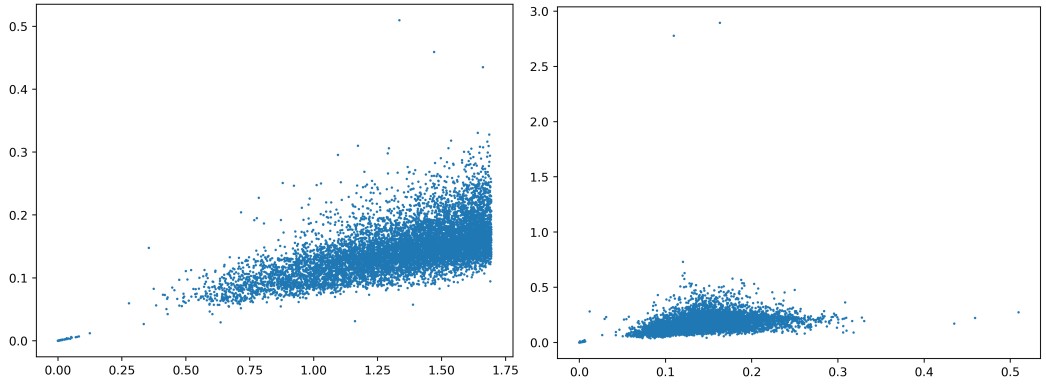

Figure 8: Distances between weaker invariants vs the complete invariant SCD for QM9 molecules. **Left**: Simplexwise Bottleneck Metric SBM_SCD vs EMD_PDD. **Right**: $L_\infty$_SDV vs SBM_SCD.

The size of the invariants SDV, PDD, SCD depends on the number $m$ of vertices in a given graph $G$. However, we can compare the stronger invariants PDD and SCD for different values of $m$. Indeed, the Earth Mover's Distance (EMD) and more general Vaserstein metrics Vaserstein (1969) work for both PDD and SCD as weighted distributions of any finite size. This comparison splits the

vertices from $V(G)$ into parts (subvertices) that are optimally 'transported' to a splitting of another vertex set $V(G')$. In our main application for molecular graphs, we compare molecules without subdividing by atoms, hence the number $m$ of atoms will be fixed for each comparison. However, to visualize all molecules from QM9 in a common space, we can consider two simplest invariants: $x$ is the average length of all edges in a graph $G$, while $y$ is the average length of all non-edges in $G$.

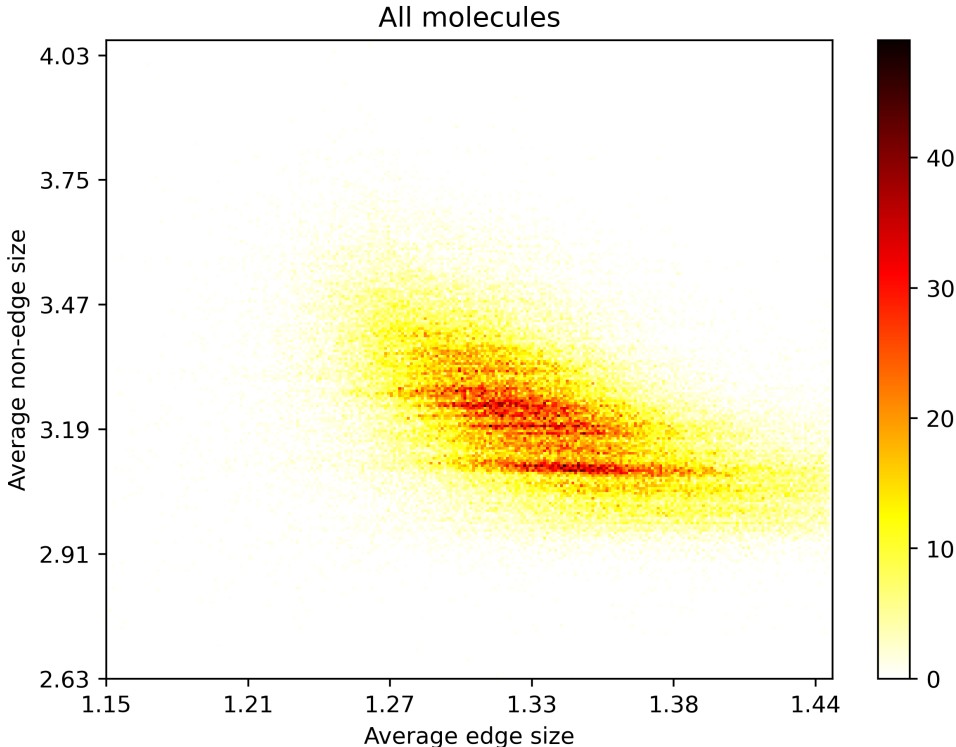

Figure 9: The visualization of all 130K+ molecular graphs in QM9 by continuous isometry invariants.

Fig. 9 shows the heat map of all 130K+ molecular graphs from QM9, where the colour of every pixel indicates the number of molecules whose invariant values $(x, y)$ fall into this pixel. The units are Angstroms, where $1\text{Å} = 10^{-10}$ meter. Fig. 9 is not a dimensionality reduction when coordinates depend on data, but is a true projection to the space of the well-defined isometry invariants $(x, y)$. Since this projection from a high-dimensional space loses some data for better visibility, stronger invariants (such as complete SCD) are needed to further explore hot spots (dark pixels).

Any black pixel in Fig. 9 defines a subgroup of about 50 molecules that have close values of the distance invariant $(x, y)$, but all these molecules can be still non-isometric and were distinguished by the SDV. Fig. 12 show the heatmaps for the largest subsets with $m = 17, 18, 19$ atoms.

## B    PROOFS OF ALL RESULTS FROM SECTION 3 FOR METRIC GRAPHS

Main classification Theorem 3.4 about the Simplexwise Distance Distributions will use two metrics, see Definitions B.3 and B.5, which also require a base metric on simpler Relative Distance Distributions in Lemma B.1 below. The $m - h$ permutable columns of the matrix $R(G; A)$ in RDD from Definition 3.3 can be interpreted as $m - h$ unordered points in $\mathbb{R}^h$. Since any isometry is bijective, the simplest metric respecting bijections is the bottleneck distance, which is also called the *Wasserstein* metric $W_\infty$. For any clouds $C, C' \subset \mathbb{R}^n$ of $m$ unordered points, the *bottleneck distance* $W_\infty(C, C') = \inf_{g:C \to C'} \sup_{p \in C} ||p - g(p)||_\infty$ is minimized over all bijections $g : C \to C'$.

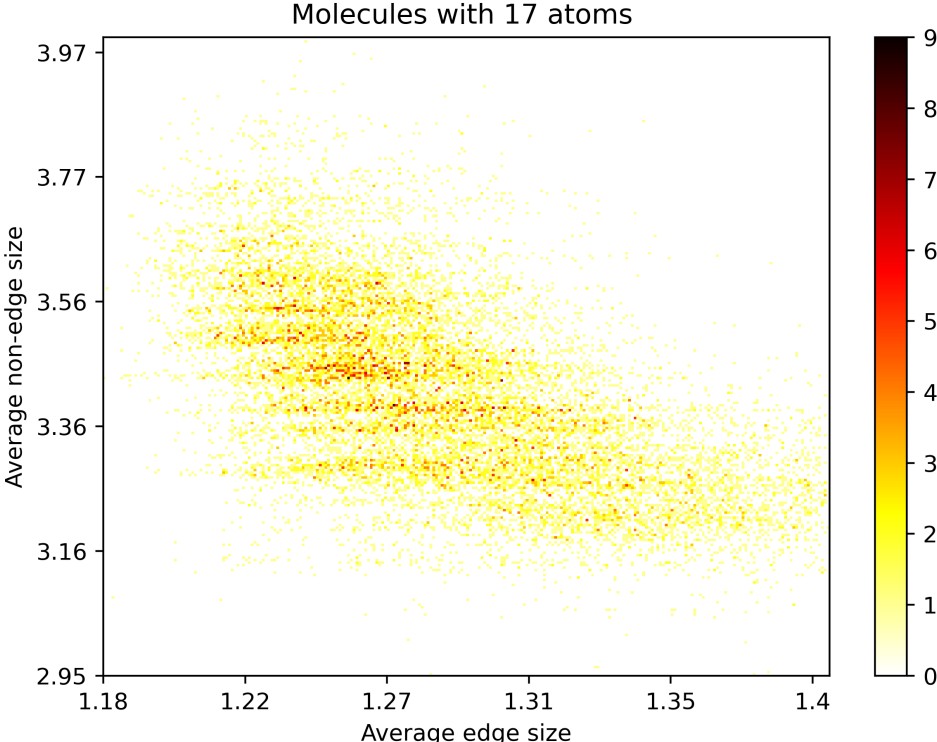

Figure 10: The heatmap of all 17394 molecular graphs with 17 atoms in the database QM9.

**Lemma B.1** (the max metric $M_\infty$ on RDDs)**.** *In any metric space, for any graphs on $m$ vertices and ordered $h$-vertex subsets $A \subset V(G)$ and $A' \subset V(G')$, set $d(\xi) = \max\{L_\infty(\xi(D(A)), D(A')), W_\infty(\xi(R(G;A)), R(G';A'))\}$ for a permutation $\xi \in S_h$ on $h$ points. Then the max metric $M_\infty(\mathrm{RDD}(G;A), \mathrm{RDD}(G';A')) = \min_{\xi \in S_h} d(\xi)$ satisfies all metric axioms on* RDD*s from Definition 3.3 and can be computed in time $O(h!(h^2 + m^{1.5}\log^h m))$.*

*Proof of Lemma B.1.* The first metric axiom says that the Relative Distance Distributions $\mathrm{RDD}(G;A)$ and $\mathrm{RDD}(G';A')$ are equivalent by Definition 3.3 if and only if $M_\infty(\mathrm{RDD}(G;A), \mathrm{RDD}(G';A')) = 0$ or $d(\xi) = 0$ for some permutation $\xi \in S_h$. Then $d(\xi) = 0$ is equivalent to $\xi(D(A)) = D(A')$ and $\xi(R(G;A)) = R(G';A')$ up to a permutation of columns due to the first axiom for the bottleneck distance $W_\infty$. The last two conclusions mean that the Relative Distance Distributions $\mathrm{RDD}(G;A), \mathrm{RDD}(G';A')$ are equivalent by Definition 3.3.

The symmetry axiom follows since any permutation $\xi$ is invertible. To prove the triangle inequality

$$M_\infty(\mathrm{RDD}(G;A), \mathrm{RDD}(G';A'))+$$
$$M_\infty(\mathrm{RDD}(G'';A''), \mathrm{RDD}(G';A')) \geq$$
$$M_\infty(\mathrm{RDD}(G;A), \mathrm{RDD}(G'';A'')),$$

let $\xi, \xi' \in S_h$ be optimal permutations for the $M_\infty$ values in the left-hand side above. The triangle inequality for $L_\infty$ says that

$$L_\infty(\xi(D(A)), D(A'))+$$
$$L_\infty(\xi'(D(A'')), D(A')) \geq$$
$$L_\infty(\xi(D(A)), \xi'(D(A''))) =$$
$$L_\infty(\xi'^{-1}\xi(D(A)), D(A'')),$$

similarly for the bottleneck distance $W_\infty$ introduced before Lemma B.1. Taking the maximum of $L_\infty, W_\infty$ preserves the triangle inequality. Then $M_\infty(\mathrm{RDD}(G;A), \mathrm{RDD}(G'';A'')) = \min_{\xi \in S_h} d(\xi)$

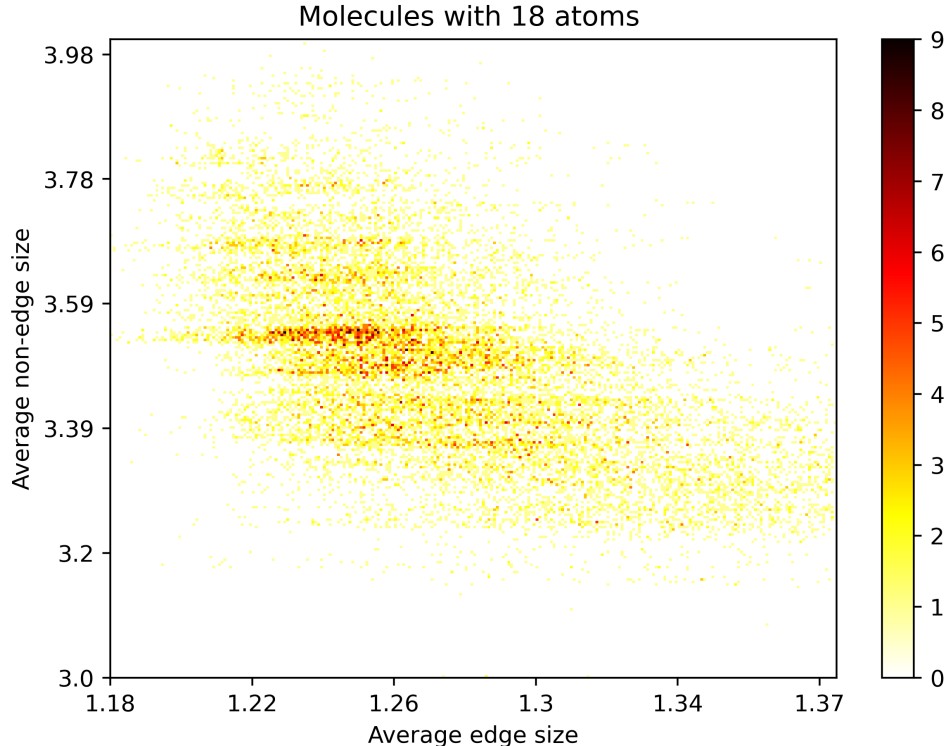

Figure 11: The heatmap of all 17836 molecular graphs with 18 atoms in the database QM9.

cannot be larger than $d(\xi'^{-1}\xi)$ for the composition of the permutations above, so the triangle inequality holds for $M_\infty$. For a fixed permutation $\xi \in S_h$, the distance $L_\infty(\xi(D(A)), D(A'))$ requires $O(h^2)$ time. The bottleneck distance $W_\infty(\xi(R(C; A)), R(C'; A'))$ on the $h \times (m - h)$ matrices $\xi(R(G; A))$ and $R(G'; A')$ with permutable columns can be considered as the bottleneck distance on clouds of $(m - h)$ unlabelled points in $\mathbb{R}^h$, so $W_\infty(\xi(R(G; A)), R(G'; A'))$ needs only $O(m^{1.5} \log^h m)$ time by (Efrat et al., 2001, Theorem 6.5). The minimization over all $\xi \in S_h$ gives the final factor $h!$. $\square$

For any metric graph $G$ on a fixed number $m$ of vertices, the Simplexwise Distance Distribution $\mathrm{SDD}(G; h)$ is an unordered collection of the same number $\binom{m}{h}$ of simpler distributions RDD.

We can define many metrics on SDDs. The simplest approach for any graphs with the same number $m$ of vertices is to apply the Simplexwise Bottleneck Metric SBM from Definition B.3.

Alternatively, if our graphs have different numbers of vertices, their SDDs can have different sizes and can become smaller after collapsing identical RDDs and assigning weights.

Any finite weighted distributions can be continuously compared by the Earth Mover's Distance EMD from Definition B.5. Firstly, we introduce the simpler metric on unordered collections of equal sizes. Definition B.2 recalls the bottleneck match for a weighted bipartite graph.

**Definition B.2** (bottleneck match $\mathrm{BM}(\Gamma)$). *Let $\Gamma$ be a complete bipartite graph with $m$ white vertices and $m$ black vertices so that every white vertex is connected to every black vertex by a single edge $e$ of a weight $w(e) \geq 0$. A* vertex matching *in $\Gamma$ is a collection $E$ of $m$ disjoint edges (with distinct vertices) of $\Gamma$. The* weight $W(E) = \max\limits_{e \in E} w(e)$ *is the largest weight of an edge in $E$. The* bottleneck match $\mathrm{BM}(\Gamma) = \min\limits_{E} W(E)$ *is the minimum weight of a vertex matching $E$ in $\Gamma$.*

Since $\Gamma$ is bipartite, any edge from a vertex matching $E$ in $\Gamma$ joins a white vertex with a black vertex. Then $\mathrm{BM}(\Gamma)$ is minimized over bijections $E$ between all white vertices and all black vertices of $G$.

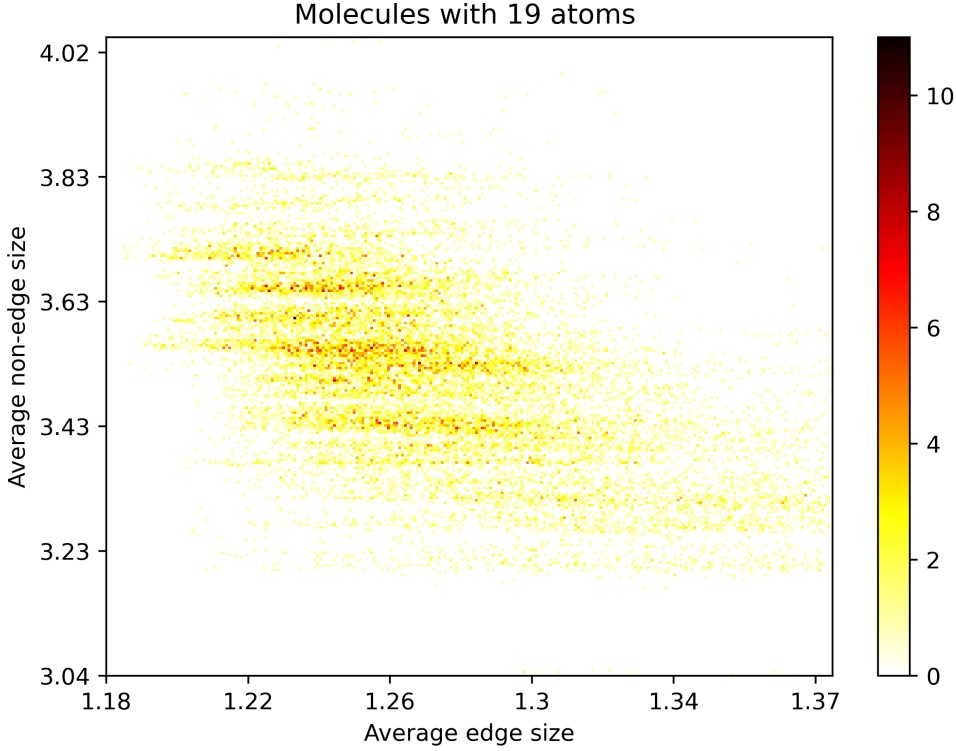

Figure 12: The heatmap of all 18336 molecular graphs with 19 atoms in the database QM9.

Definition B.3 builds a weighted bipartite graph $\Gamma(G, G')$ on all Relative Distance Distributions (RDDs) of given graphs $G, G'$ and then introduces the Simplexwise Bottleneck Metric on SDDs of $G, G'$ as the bottleneck match BM of $\Gamma(G, G')$ with weights defined by the metric $M_\infty$ on RDDs.

**Definition B.3** (SBM : Simplexwise Bottleneck Metric on SDDs). *Let $h \geq 1$ and $G, G'$ be any graphs on $m$ unordered vertices in a metric space. For each graph, consider one of $\binom{m}{h}$ unordered subsets of $h$ vertices in $G, G'$, which form arbitrarily ordered sequences $A, A'$, respectively. The complete bipartite graph $\Gamma(G, G')$ has $\binom{m}{h}$ white vertices and $\binom{m}{h}$ black vertices. Then any edge $e$ of $\Gamma(G, G')$ has endpoints associated with $\mathrm{RDD}(G; A)$ and $\mathrm{RDD}(G'; A')$, and the weight defined as $w(e) = M_\infty(\,\mathrm{RDD}(G; A),\ \mathrm{RDD}(G'; A')\,)$. The* Simplexwise Bottleneck Metric $\mathrm{SBM}(\mathrm{SDD}(G; h), \mathrm{SDD}(G'; h)) = \mathrm{BM}(\Gamma(G, G'))$ *is the bottleneck match of $\Gamma(G, G')$.*

**Lemma B.4** (metric axioms for SBM on SDDs). *The Simplexwise Bottleneck Metric $\mathrm{SBM}(S, Q)$ from Definition B.3 satisfies all metric axioms on Simplexwise Distance Distributions $S, Q$.*

*Proof of Lemma B.4.* The coincidence axiom means that $\mathrm{SBM}(S, Q) = 0$ if and only if the unordered distributions $S, Q$ are equal. Indeed, if $S, Q$ can be matched by a permutation, we get a vertex matching $E$ of $\Gamma(S, Q)$ whose all edges have weights $w(e) = 0$ Definitions B.2 and B.3 imply that $\mathrm{SBM}(S, Q) = \mathrm{BM}(\Gamma(S, Q)) = 0$. Conversely, if $\mathrm{SBM}(S, Q) = \mathrm{BM}(\Gamma(S, Q)) = 0$, there is a vertex matching $E$ in $\Gamma(S, Q)$ with all $w(e) = 0$. This matching $E$ defines a required bijection $S \to Q$. The symmetry axiom $\mathrm{SBM}(S, Q) = \mathrm{SBM}(Q, S)$ follows from Definition B.3 and the symmetry of $M_\infty$ in Lemma B.1.

To prove the triangle inequality $\mathrm{SBM}(S, Q) + \mathrm{SBM}(Q, T) \geq \mathrm{SBM}(S, T)$, let $E_{SQ}, E_{QT}$ be optimal vertex matchings in the graphs $\Gamma(S, Q), \Gamma(Q, T)$, respectively, such that $\mathrm{SBM}(S, Q) = W(E_{SQ})$ and $\mathrm{SBM}(Q, T) = W(E_{QT})$, see Definition B.2. The composition $E_{SQ} \circ E_{QT}$ is a vertex matching in $\Gamma(S, T)$, so $W(E_{SQ} \circ E_{QT}) \geq \mathrm{SBM}(S, T)$. It suffices to prove that

$$W(E_{SQ}) + W(E_{QT}) \geq W(E_{SQ} \circ E_{QT}).$$

Let $e_{ST}$ be an edge with a largest weight from $E_{SQ} \circ E_{QT}$, so $W(E_{SQ} \circ E_{QT}) = w(e_{ST})$. The edge $e_{ST}$ can be considered the union of edges $e_{SQ} \in E_{SQ}$ and $e_{QT} \in E_{QT}$. By the triangle inequality for $M_\infty$ from Lemma B.1: $w(e_{SQ}) + w(e_{QT}) \geq w(e_{ST}) = W(E_{SQ} \circ E_{QT})$ implies that $W(E_{SQ}) + W(E_{QT}) \geq W(E_{SQ} \circ E_{QT})$ because both terms on the left-hand side are maximized for all edges (not only $e_{SQ}, e_{QT}$) from $E_{SQ}, E_{QT}$. $\qquad\square$

Definition B.5 below makes sense for any weighted distributions of the form $\{[w_1, R_1], \ldots, [w_m, R_m]\}$, where $R_1, \ldots, R_m$ are objects with a base metric $\mu$ and weights $w_1, dots, w_m \in [0, 1]$, respectively, satisfying $\sum_{i=1}^{m} w_i = 1$. For example, objects $R_i$ can be the rows of the matrix PDD (Pointwise Distance Distribution) with the base metric $L_\infty(u, v) = |u - v|_\infty$ equal to the maximum absolute difference between corresponding coordinates of vectors $u, v$.

**Definition B.5** (EMD). *Let $S = \{[w_i(S), R_i(S)]\}_{i=1}^{m(S)}$ and $Q = \{[w_i(Q), R_i(Q)]\}_{i=1}^{m(Q)}$ be finite weighted distributions. A flow from $S$ to $Q$ is an $m(S) \times m(Q)$ matrix whose element $f_{ij} \in [0, 1]$ represents a partial flow from $R_i(S)$ to $R_j(Q)$. The Earth Mover's Distance is the minimum cost $\mathrm{EMD}(S, Q) = \sum_{i=1}^{m(S)} \sum_{j=1}^{m(Q)} f_{ij}\mu(R_i(S), R_j(Q))$ for $f_{ij} \in [0, 1]$ subject to $\sum_{j=1}^{m(Q)} f_{ij} \leq w_i(S)$ for $i = 1, \ldots, m(S)$, $\sum_{i=1}^{m(S)} f_{ij} \leq w_j(Q)$ for $j = 1, \ldots, m(Q)$, and $\sum_{i=1}^{m(S)} \sum_{j=1}^{m(Q)} f_{ij} = 1$.*

The first condition $\sum_{j=1}^{m(Q)} f_{ij} \leq w_i(S)$ means that not more than the weight $w_i(S)$ of the component $R_i(S)$ 'flows' into all components $R_j(Q)$ via 'flows' $f_{ij}$, $j = 1, \ldots, m(Q)$. Similarly, the second condition $\sum_{i=1}^{m(S)} f_{ij} = w_j(Q)$ means that all 'flows' $f_{ij}$ from $R_i(S)$ for $i = 1, \ldots, m(S)$ 'flow' into $R_j(Q)$ up to the maximum weight $w_j(Q)$. The last condition $\sum_{i=1}^{m(S)} \sum_{j=1}^{m(Q)} f_{ij} = 1$ forces to 'flow' all rows $R_i(S)$ to all rows $R_j(Q)$. The EMD satisfies all metric axioms, see the appendix in Rubner et al. (2000), needs $O(m^3 \log m)$ time for distributions of a maximum size $m$ and is approximated in $O(m)$ time, see Shirdhonkar & Jacobs (2008); Sato et al. (2020).

We split the proof of Theorem 3.4 into the four shorter parts below:

(a) the invariance and time complexity of the Simpexwise Distance Distribution $\mathrm{SDD}(G; h)$;

(b) metric axioms and the time complexity for the Simplexwise Bottleneck Metric (SBM) on SDDs;

(c) metric axioms and the time complexity for the Earth Mover's Distance (EMD) on SDDs;

(d) the Lipschitz continuity of SDD in SBM and EMD.

*Proof of Theorem 3.4(a).* Any isometry $G \to F$ preserves distances, hence induces a bijection $\mathrm{SDD}(G; h) \to \mathrm{SDD}(F; h)$ for $h \geq 1$.

By Definition 3.3, for any $h \geq 1$ and a graph $G$ on $m$ unlabelled vertices in a metric space, the Simplexwise Distance Distribution $\mathrm{SDD}(G; h)$ of consists of $\binom{m}{h} = \frac{m!}{h!(m-h)!} = O(m^h/h!)$ Relative Distance Distributions $\mathrm{RDD}(G; A)$ for any unordered subset $A \subset V(G)$ of $h$ vertices.

For any order of the vertices in $A$, every $\mathrm{RDD}(G; A)$ consists of the distance matrix $D(A)$, which needs $O(h^2)$ time and $h \times (m - h)$ matrix $R(G; A)$, which needs $h(m - h)$ time. Since $h \leq m$, the extra factor $O(hm)$ gives the final time $O(m^{h+1}/(h-1)!)$ for $\mathrm{SDD}(G; h)$. $\qquad\square$

*Proof of Theorem 3.4(b).* The complete bipartite graph $\Gamma(G, G')$ in Definition B.3 has $V = 2\binom{m}{h} = O\left(\frac{m^h}{h!}\right)$ vertices and $E = O\left(\frac{m^{2h}}{(h!)^2}\right)$ edges. The weight $w(e)$ of each edge $e$ equal the metric $M_\infty$, which needs time $O((h^2 + m^{1.5} \log^h m)h!)$ by Lemma B.1. Since $h$ is much smaller than $m$, we can assume that $h^2 \leq O(m^{1.5} \log^h m)$ and drop $h^2$ in further complexities. So we

computed the full graph $\Gamma(G, G')$ in time $O\left(\dfrac{m^{2h+1.5}}{h!} \log^h m\right)$. After that, the bottleneck match $\mathrm{BM}(\Gamma(G, G'))$ can be computed by Hopcroft & Karp (1973) in time $O(E\sqrt{V}) = O\left(\dfrac{m^{2.5h}}{(h!)^{2.5}}\right)$. The total time for $\mathrm{SBM}(\mathrm{SDD}(G; h), \mathrm{SDD}(G'; h)) = \mathrm{BM}(\Gamma(G, G'))$ in Definition B.3 has the upper bound $O\left(\dfrac{m^{2h+1.5}}{h!} \log^h m\right) + O\left(\dfrac{m^{2.5h}}{(h!)^{2.5}}\right) = O\left(\dfrac{m^{2.5h} + m^{2h+1.5} \log^h m}{h!}\right)$. $\qquad\square$

*Proof of Theorem 3.4(c).* The metric axioms for the Earth Mover's Distance (EMD) are proved in the appendix of Rubner et al. (2000) assuming the metric axioms for the underlying distance $d$, which is the metric $M_\infty$ from Lemma C.4 in our case. The time $O(h!(h^2 + m^{1.5} \log^h m)l^2 + l^3 \log l)$ for EMD of a maximum size $l$ follow from the time $O((h^2 + m^{1.5} \log^h m)h!)$ for $M_\infty$ in Lemma B.1, after multiplying by a quadratic factor for the size of cost matrices and adding near-cubic time, see Fredman & Tarjan (1987); Goldberg & Tarjan (1987). $\qquad\square$

The Lipschitz continuity of SDD in Theorem 3.4(d) needs Lemma B.6 for simpler distributions.

**Lemma B.6** (Lipschitz continuity of $\mathrm{RDD}(G; A)$)**.** *Let $A$ be an ordered sequence of $h \geq 1$ vertices in a graph $G$ in a metric space. Let $A'$ and $G'$ be obtained from $A$ and $G$, respectively, by perturbing every vertex in its $\varepsilon$-neighborhood. Then the Relative Distance Distribution changes in the metric from Lemma B.1 by at most $2\varepsilon$, so $M_\infty(\mathrm{RDD}(G; A), \mathrm{RDD}(G'; A')) \leq 2\varepsilon$.*

*Proof.* Order all vertices of the given graphs $G, G'$ so that every vertex $p_i \in V(G)$ has the same index as its perturbation $p_i' \in V(G')$. In the given metric space, the $G$-based distance $d_G(p_i, p_j)$ between any vertices in $V(G)$ changes under perturbation by at most $2\varepsilon$ so that $|d_G(p_i, p_j) - d_G(p_i', p_j')| \leq 2\varepsilon$, which also holds for negative distances of non-edges. The 1-1 correspondence $p_i \leftrightarrow p_i'$ induces the (trivial) permutation $\xi$ on $h$ vertices and another (trivial) permutation on $m - h$ columns of the matrix $R(G; A)$. For these fixed permutations, both distances $L_\infty(\xi(D(A)), D(A'))$ and $W_\infty(\xi(R(G; A)), R(G', A'))$ in Lemma B.1 have the upper bound $2\varepsilon$ because all corresponding elements in the underlying matrices change by at most $2\varepsilon$. Taking the minimum for all permutations in Lemma B.1 gives the required upper bound of $2\varepsilon$ for $M_\infty$. $\qquad\square$

Theorem 3.4(d) substantially generalizes the fact that perturbing two points in their $\varepsilon$-neighborhoods changes the Euclidean distance between these points by at most $2\varepsilon$.

*Proof of Theorem 3.4(d).* The upper bound of Lemma B.6 extends to SBM in Definition B.3 and EMD in Definition B.5 because all distances change by at $2\varepsilon$ and the total weight of all flows is 1, so $\mathrm{SBM}(\mathrm{SDD}(G; h), \mathrm{SDD}(G'; h)) \leq 2\varepsilon$ and $\mathrm{EMD}(\mathrm{SDD}(G; h), \mathrm{SDD}(G'; h)) \leq 2\varepsilon$. $\qquad\square$

## C  PROOFS OF ALL RESULTS FROM SECTION 4 FOR EUCLIDEAN GRAPHS

The comprehensive book "Euclidean Distance Geometry" (Liberti & Lavor, 2017, Chapter 3) discusses realizations in $\mathbb{R}^n$ of a complete graph with ordered vertices given by a full distance matrix.

The case of unordered vertices is much harder. Lemma C.2(a) and later results hold for all graphs including degenerate ones, for example, when three vertices points are in a straight line.

The *affine dimension* $0 \leq \mathrm{aff}(A) \leq n$ of a point cloud $A = \{p_1, \ldots, p_m\} \subset \mathbb{R}^n$ is the maximum dimension of the vector space generated by all inter-point vectors $p_i - p_j$, $i, j \in \{1, \ldots, m\}$. Then $\mathrm{aff}(A)$ is an isometry invariant and is independent of an order of points of $A$. Any cloud $A$ of 2 distinct points has $\mathrm{aff}(A) = 1$. Any cloud $A$ of 3 points that are not in the same straight line has $\mathrm{aff}(A) = 2$. Any $n - 1$ points of $A$ have $\mathrm{aff} \leq n - 2$. For example, any two distinct points in $A \subset \mathbb{R}^3$ generate a straight line.

Lemma C.2(c) proves that $\mathrm{OCD}(G; A)$ suffices to reconstruct any Euclidean graph $G \subset \mathbb{R}^n$ with the center of mass at the origin $0 \in \mathbb{R}^n$ for a suitable ordered subset $A \subset V(G)$ of $n - 1$ vertices. In $\mathbb{R}^2$, any point $p_1 \neq O(A)$ forms a suitable 1-vertex set $A$. In $\mathbb{R}^3$, one can choose any distinct vertices $p_1, p_2 \in V(G)$ so that the infinite straight line through $p_1, p_2$ avoids $0 \in \mathbb{R}^n$. If there are

no such $p_1, p_2$, then $G \subset \mathbb{R}^3$ is contained in a straight line $L$, so $\mathrm{aff}(V(G)) = 1$. In this degenerate case, the stronger condition $\mathrm{aff}(V(G) \cup \{0\}) = \mathrm{aff}(V(G))$ will help reconstruct $G \subset L$ by using any point $p_1 \neq 0$.

The first step is to reconstruct any ordered sequence from its square distance matrix in Lemma C.2(a). Lemma C.1 provides a simple criterion for a matrix to be realizable by squared distances of a point cloud in $\mathbb{R}^n$.

**Lemma C.1** (realization of distances). *A symmetric $m \times m$ matrix of $s_{ij} \geq 0$ with $s_{ii} = 0$ is realizable as a matrix of squared distances between points $p_0 = 0, p_1, \ldots, p_{m-1} \in \mathbb{R}^n$ if and only if the $(m-1) \times (m-1)$ matrix $g_{ij} = \dfrac{s_{0i} + s_{0j} - s_{ij}}{2}$ has only non-negative eigenvalues and $k \leq n$ of them are positive. If yes, then $g_{ij} = p_i \cdot p_j$ is the Gram matrix of $p_1, \ldots, p_{m-1}$, which are uniquely determined in time $O(m^3)$ up to an orthogonal map in $\mathbb{R}^n$.*

*Proof of Lemma C.1.* We extend (Dekster & Wilker, 1987, Theorem 1) to the case $m \geq n + 1$ and give a reference for the time $O(m^3)$ to determine the points $p_1, \ldots, p_{m-1}$ up to isometry in $\mathbb{R}^n$.

The part *only if* $\Rightarrow$. Let a symmetric matrix $S$ consist of squared distances between points $p_0 = 0, p_1, \ldots, p_{m-1} \in \mathbb{R}^n$. For $i, j = 1, \ldots, m-1$, the matrix $G$ with the elements

$$g_{ij} = \frac{s_{0i} + s_{0j} - s_{ij}}{2} = \frac{p_i^2 + p_j^2 - |p_i - p_j|^2}{2} = p_i \cdot p_j$$

is the Gram matrix, which can be written as $G = P^T P$, where the columns of the $n \times (m-1)$ matrix $P$ are the vectors $p_1, \ldots, p_{m-1}$. For any vector $v \in \mathbb{R}^{m-1}$, we have

$$0 \leq |Pv|^2 = (Pv)^T (Pv) = v^T (P^T P) v = v^T G v.$$

Since the quadratic form $v^T G v \geq 0$ for any $v \in \mathbb{R}^{m-1}$, the matrix $G$ is positive semi-definite meaning that (Horn & Johnson, 2012, Theorem 7.2.7) $G$ has only non-negative eigenvalues.

The part *if* $\Leftarrow$. For any positive semi-definite matrix $G$, there is an orthogonal matrix $Q$ such that $Q^T G Q = D$ is the diagonal matrix, whose $m - 1$ diagonal elements are non-negative eigenvalues of $G$. The diagonal matrix $\sqrt{D}$ consists of the square roots of eigenvalues of $D$ on the diagonal. The number of positive eigenvalues of $G$ equals the dimension $k = \mathrm{aff}(\{p_0, \ldots, p_{m-1}\})$ of the subspace in $\mathbb{R}^n$ linearly spanned by $p_1, \ldots, p_{m-1}$. We may assume that all $k \leq n$ positive eigenvalues of $G$ correspond to the first $k$ coordinates of $\mathbb{R}^n$. Since $Q^T = Q^{-1}$, the given matrix $G = QDQ^T = (Q\sqrt{D})(Q\sqrt{D})^T$ becomes the Gram matrix of the columns of $Q\sqrt{D}$. After removing the last $m - n - 1$ zero coordinates, these columns become vectors $p_1, \ldots, p_{m-1} \in \mathbb{R}^n$, which are uniquely determined up to an orthogonal map. Computing $p_1, \ldots, p_{m-1}$ requires a diagonalization (Press et al., 2007, section 11.5) of $G$ in time $O(m^3)$. $\square$

Lemma C.2 extends (Widdowson & Kurlin, 2023, Lemma E.5) by proving a time for a point cloud reconstruction based on Lemma C.1.

**Lemma C.2** (reconstruction). *(a) Any sequence of ordered points $A = (p_1, \ldots, p_m)$ in $\mathbb{R}^n$ can be reconstructed (uniquely up to isometry) from the matrix of the Euclidean distances $|p_i - p_j|$ in time $O(m^3)$. If all distances are divided by $R = \max\limits_{i=1,\ldots,m} |p_i|$, the reconstruction of $A \subset \mathbb{R}^n$ is unique up to isometry and uniform scaling in $\mathbb{R}^n$.*

*(b) If $m \leq n$, the uniqueness of reconstructions in part (a) remains true if we replace isometry by rigid motion in $\mathbb{R}^n$.*

*(c) Any Euclidean graph $G \subset \mathbb{R}^n$ on $m$ unlabeled vertices whose center of mass is the origin $0 \in \mathbb{R}^n$ can be reconstructed (uniquely up to orientation-preserving rotations of $\mathbb{R}^n$ around $0$) from the Oriented Centered Distribution $\mathrm{OCD}(G; A)$ in time $O(m^3)$ for any ordered subset $A \subset V(G)$ of $n - 1$ vertices with $\mathrm{aff}(A \cup \{0\}) = \mathrm{aff}(V(G))$. If $\mathrm{aff}(V(G)) = n$, then $\mathrm{aff}(A \cup \{0\}) = n - 1$ suffices. The vertex set $V(G)$ has a suitable subset $A \subset V(G)$ in all cases.*

*Proof of Lemma C.2.* (a) By translation, we can put the first point $p_1$ at the origin $0 \in \mathbb{R}^n$. Let $\Gamma$ be the $(m-1) \times (m-1)$ matrix $g_{ij} = \dfrac{p_i^2 + p_j^2 - |p_i - p_j|^2}{2} = p_i \cdot p_j$ constructed from squared distances

between $p_1 = 0, \ldots, p_m$ for $i, j = 2, \ldots, m$. By Lemma C.1 if $\Gamma$ has $k \leq n$ positive eigenvalues, then $p_1 = 0, \ldots, p_m$ can be uniquely determined up to isometry in $\mathbb{R}^k \subset \mathbb{R}^n$ in time $O(m^3)$. If all distances are divided by the same radius $R$, the above construction guarantees uniqueness up to isometry and uniform scaling.

**(b)** If $m \leq n$, any mirror images of $A \subset \mathbb{R}^n$ after a suitable rigid motion in $\mathbb{R}^n$ can be assumed to belong to an $(n-1)$-dimensional hyperspace $H \subset \mathbb{R}^n$, where they are matched by a mirror reflection $H \to H$ with respect to an $(n-2)$-dimensional subspace $S \subset H$, which is realized by the $180°$ orientation-preserving rotation of $\mathbb{R}^n$ around $S$.

**(c)** We will reconstruct a Euclidean graph $G \subset \mathbb{R}^n$ from $\mathrm{OCD}(G; A \cup \{0\})$ so that the center of mass of $V(G)$ is the origin $0 \in \mathbb{R}^n$. If $\mathrm{aff}(V(G)) = k < n$, the vertex set $V(G) \subset \mathbb{R}^n$ is contained in a $k$-dimensional subspace, which can be assumed (up to rotation of $\mathbb{R}^n$) to be the subspace $\mathbb{R}^k \subset \mathbb{R}^n$ for the first $k$ of $n$ coordinates in $\mathbb{R}^n$. Since $\mathrm{aff}(A \cup \{0\}) = k$, some $k$ vectors from the origin $0$ to (say) the first vertices $p_1, \ldots, p_k$ of $A$ form a linear basis of $\mathbb{R}^k$. The $k$ vertices $p_1, \ldots, p_k$ are uniquely reconstructed up to rotations around the origin $0$ in $\mathbb{R}^k$ from $D(A \cup \{0\})$ by part (b). Any other vertex $q \in V(G) - \{p_1, \ldots, p_k\}$ is uniquely determined by its column the matrix $M(G; A \cup \{0\})$ in Definition 4.1 as a unique intersection of the $k + 1$ spheres $S(p_i; d(q, p_i))$ with the centers $p_i$ and radii $d(q, p_i)$ for $i = 0, \ldots, k$ and $p_0 = 0$,

In the generic case $\mathrm{aff}(V(G)) = n$, the condition $\mathrm{aff}(A \cup \{0\}) = n - 1$ means that the vectors $p_1, \ldots, p_{n-1}$ from the origin $0$ to the $n - 1$ vertices of $A$ are linearly independent. The sequence $(p_1, \ldots, p_{n-1}, 0)$ of $n$ points including the origin $0$ can be uniquely reconstructed from $D(A \cup \{0\})$ up to rigid motion in $\mathbb{R}^n$ by part (b).

Any other vertex $q \in V(G) - A$ is uniquely determined by its column in $M(G; A \cup \{0\})$ as follows. The $n$ spheres $S(p_i; d(q, p_i))$ with the centers $p_i$ and radii $d(q, p_i)$ for $i = 0, \ldots, n - 1$ and $p_0 = 0$, contain $q$ and intersect in one or two points. We can uniquely choose $q$ among these two options due to the sign of the determinant (in the bottom row of $\mathrm{OCD}(G; A)$) of the vectors $q - p_0, \ldots, q - p_{n-1}$.

The presence (or absence) of an edge between any $p, q \in V(G)$ is determined by the sign $+1$ (or $-1$, respectively) of each distance $d_G(p, q)$ in the matrices $D(A \cup \{0\})$ and $M(G; A \cup \{0\})$.  $\square$

Lemma C.2(b) for $m = n = 3$ implies that any triangle is determined by its sides up to rigid motion in $\mathbb{R}^3$. For example, the sides $3, 4, 5$ define a right-angled triangle whose mirror images are not related by rigid motion inside a plane $H \subset \mathbb{R}^3$, but are matched by composing a suitable rigid motion in $H$ and a $180°$ rotation of $\mathbb{R}^3$ around a line in $H$.

Theorem C.3 says that the strength $\sigma(A)$ of a simplex is roughly 'linear' in point coordinates of $A$.

**Theorem C.3** (Lipschitz continuity of $\sigma$, proved in (Widdowson & Kurlin, 2023, Theorem 4.4)). *Let a cloud $A'$ be obtained from another $(n + 1)$-point cloud $A \subset \mathbb{R}^n$ by perturbing every point within its $\varepsilon$-neighborhood. Then $|\sigma(A') - \sigma(A)| \leq 2\varepsilon c_n$ for a constant $c_n$, where $c_2 = 2\sqrt{3}$, $c_3 \approx 0.43$.*

The strength $\sigma(A)$ from Definition 4.2 will take care of extra signs in OCDs and allows us to prove the analog of Lemma B.1 for a similar time complexity with $h = n$.

Theorem 4.4 needs the Lipschitz continuity of $s\sigma(A)$, when a sign $s \in \{\pm 1\}$ from a bottom row of OCD discontinuously changes while passing through a degenerate set $A$. The usual volume of $A$ is not Lipschitz continuous: consider the triangle with two vertices fixed at $(\pm l, 0)$ and one moving vertex $(0, t\varepsilon)$ for $t \in [-1, 1]$. The signed area of the triangle significantly changes from $-l\varepsilon$ (unbounded because $l$ can be large for any given small $\varepsilon$) to $0$ (when $t = 0$ and the triangle degenerates to 3 points in a line), then to $l\varepsilon$ (when $t = 1$). The change of the signed area is $2l\varepsilon$ while only one vertex is shifted by $2\varepsilon$, so the Lipschitz constant in this example can be as large as half-distance between given points.

**Lemma C.4** (metric on OCDs). *Using the strength $\sigma$ from Definition 4.2, we consider the bottleneck distance $W_\infty$ on the set of permutable $m - n + 1$ columns of $M(G; A \cup \{0\})$ as on the set of $m - n + 1$ unordered points $\left( v, \dfrac{s}{c_n} \sigma(A \cup \{0, q\}) \right) \in \mathbb{R}^{n+1}$. For another $\mathrm{OCD}' = [D(A' \cup \{0\}); M(G'; A' \cup \{0\})]$ and any permutation $\xi \in S_{n-1}$ of indices $1, \ldots, n - 1$ acting on $D(A \cup \{0\})$ and the first*

$n - 1$ *rows of* $M(G; A \cup \{0\})$, *set* $d_o(\xi) = \max\{L, W\}$, *where*

$$L = L_\infty\Big(\xi(D(A \cup \{0\})), D(A' \cup \{0\})\Big), \quad W = W_\infty\Big(\xi(M(G; A \cup \{0\})), M(G'; A' \cup \{0\})\Big).$$

*Then* $M_\infty(\text{OCD}, \text{OCD}') = \min\limits_{\xi \in S_{n-1}} d_o(\xi)$ *satisfies all metric axioms on Oriented Centered Distributions* (OCD*s*) *and can be computed in time* $O((n-1)!(n^2 + m^{1.5} \log^n m))$.

The coefficient $\dfrac{1}{c_n}$ in front of $\sigma(A \cup \{q\})$ normalizes the Lipschitz constant $c_n$ of $\sigma$ to 1 in line with changes of distances by at most $2\varepsilon$ when points are perturbed within their $\varepsilon$-neighborhoods.

*Proof of Lemma C.4.* The first metric axiom says that $\text{OCD}(G; A), \text{OCD}(G'; A')$ are equivalent by Definition 4.1 if and only if $M_\infty(\text{OCD}(G; A), \text{OCD}(G'; A')) = 0$ or $d(\xi) = 0$ for some permutation $\xi \in S_n$. Then $d(\xi) = 0$ is equivalent to $\xi(D(A)) = D(A')$ and $\xi(M(G; A)) = M(G'; A')$ up to a permutation of columns due to the first axiom for $W_\infty$. The last two conclusions mean that the Oriented Centered Distributions $\text{OCD}(G; A), \text{OCD}(G'; A')$ are equivalent by Definition 4.1. The symmetry axiom follows since any permutation $\xi$ is invertible. To prove the triangle inequality

$$M_\infty(\text{OCD}(G; A), \text{OCD}(G'; A')) +$$
$$M_\infty(\text{OCD}(G''; A''), \text{OCD}(G'; A')) \geq$$
$$M_\infty(\text{OCD}(G; A), \text{OCD}(G''; A'')),$$

let $\xi, \xi' \in S_h$ be optimal permutations for the $M_\infty$ values in the left-hand side above. The triangle inequality for $L_\infty$ says that

$$L_\infty(\xi(D(A)), D(A')) +$$
$$L_\infty(\xi'(D(A'')), D(A')) \geq$$
$$L_\infty(\xi(D(A)), \xi'(D(A''))) =$$
$$L_\infty(\xi'^{-1}\xi(D(A)), D(A'')),$$

similarly for the bottleneck distance $W_\infty$. Taking the maximum of the metrics $L_\infty, W_\infty$ preserves the triangle inequality. Then $M_\infty(\text{OCD}(C; A), \text{OCD}(C''; A'')) = \min\limits_{\xi \in S_n} d(\xi)$ cannot be larger than $d(\xi'^{-1}\xi)$ for the composition of the permutations above, so the triangle inequality holds for $M_\infty$.

For a fixed permutation $\xi \in S_{n-1}$, the distance $L_\infty(\xi(D(A)), D(A'))$ requires $O(n^2)$ time. The bottleneck distance $W_\infty(\xi(M(G; A)), M(G'; A'))$ on the $(n+1) \times (m-h)$ matrices $\xi(M(C; A))$ and $M(C'; A')$ with permutable columns can be considered the bottleneck distance on clouds of $(m-h)$ unlabeled points in $\mathbb{R}^h$, so $W_\infty(\xi(M(G; A)), M(G'; A'))$ needs only $O(m^{1.5} \log^h m)$ time by (Efrat et al., 2001, Theorem 6.5). The minimization over all $\xi \in S_n$ gives the factor $n!$

When computing the metric $M_\infty$ on OCDs, each of the $m - n$ signs for a vertex $q \in V(G) - A$ is multiplied by the strength $\sigma(A)$. The strength $\sigma(A)$ in Definition 4.2 is computed from all pairwise distances in $D(A)$ via the Cayley-Menger determinant Sippl & Scheraga (1986).

One $(n+2) \times (n+2)$ determinant needs time $O(n^3)$ by Gaussian elimination. All $m - n$ strengths need $O(mn^3)$ time, which is smaller than the later time including the factors $m^{1.5}$ and $n!$ Then $m - n$ permutable columns of $M(G; A)$ are considered as $m - n$ unlabelled points in $\mathbb{R}^{n+1}$, which explains the extra factor $\log m$ coming from (Efrat et al., 2001, Theorem 6.5). □

For any Euclidean graph $G \subset \mathbb{R}^n$ on a fixed number $m$ of vertices, the Simplexwise Centered Distribution $\text{SCD}(G)$ is an unordered collection of the same number $\binom{m}{n-1}$ of simpler distributions OCD. Similarly to SDD of metric graphs, we apply the Simplexwise Bottleneck Metric SBM from Definition C.5 to SCDs for any graphs with the same number $m$ of vertices. The Earth Mover's Distance EMD from Definition B.5 allows us to continuously compare SCDs of different sizes, which can become smaller after collapsing identical OCDs and assigning weights.

**Definition C.5** (SBM : Simplexwise Bottleneck Metric on SCDs). *Let* $G, G'$ *be any graphs on* $m$ *unordered vertices in* $\mathbb{R}^n$ *with centers of mass at the origin. For each graph, consider one of* $\binom{m}{n-1}$ *unordered subsets of* $n-1$ *vertices in* $G, G'$, *which form arbitrarily ordered sequences* $A, A'$, *respectively. The complete bipartite graph* $\Gamma(G, G')$ *has* $k = \binom{m}{n-1}$ *white vertices and* $k$ *black vertices.*

*Then any edge $e$ of $\Gamma(G, G')$ has endpoints associated with $\mathrm{RDD}(G; A)$ and $\mathrm{RDD}(G'; A')$, and the* weight *defined as $w(e) = M_\infty\big( \mathrm{OCD}(G; A), \mathrm{OCD}(G'; A') \big)$. The* Simplexwise Bottleneck Metric $\mathrm{SBM}(\mathrm{SCD}(G), \mathrm{SCD}(G')) = \mathrm{BM}(\Gamma(G, G'))$ *is the bottleneck match of $\Gamma(G, G')$.*

**Lemma C.6** (metric axioms for SBM on SCDs)**.** *The Simplexwise Bottleneck Metric $\mathrm{SBM}(S, Q)$ from Definition C.5 satisfies all metric axioms on Simplexwise Centered Distributions $S, Q$.*

*Proof of Lemma C.6.* The coincidence axiom means that $\mathrm{SBM}(S, Q) = 0$ if and only if the unordered distributions $S, Q$ are equal. Indeed, if $S, Q$ can be matched by a permutation, we get a vertex matching $E$ of $\Gamma(S, Q)$ whose all edges have weights $w(e) = 0$ Definitions B.2 and C.5 imply that $\mathrm{SBM}(S, Q) = \mathrm{BM}(\Gamma(S, Q)) = 0$. Conversely, if $\mathrm{SBM}(S, Q) = \mathrm{BM}(\Gamma(S, Q)) = 0$, there is a vertex matching $E$ in $\Gamma(S, Q)$ with all $w(e) = 0$. This matching $E$ defines a required bijection $S \to Q$. The symmetry axiom $\mathrm{SBM}(S, Q) = \mathrm{SBM}(Q, S)$ follows from Definition C.5 and the symmetry of $M_\infty$ in Lemma C.4.

To prove the triangle inequality $\mathrm{SBM}(S, Q) + \mathrm{SBM}(Q, T) \geq \mathrm{SBM}(S, T)$, let $E_{SQ}, E_{QT}$ be optimal vertex matchings in the graphs $\Gamma(S, Q), \Gamma(Q, T)$, respectively, such that $\mathrm{SBM}(S, Q) = W(E_{SQ})$ and $\mathrm{SBM}(Q, T) = W(E_{QT})$, see Definition B.2. The composition $E_{SQ} \circ E_{QT}$ is a vertex matching in $\Gamma(S, T)$, so $W(E_{SQ} \circ E_{QT}) \geq \mathrm{SBM}(S, T)$. It suffices to prove that

$$W(E_{SQ}) + W(E_{QT}) \geq W(E_{SQ} \circ E_{QT}).$$

Let $e_{ST}$ be an edge with a largest weight from $E_{SQ} \circ E_{QT}$, so $W(E_{SQ} \circ E_{QT}) = w(e_{ST})$. The edge $e_{ST}$ can be considered the union of edges $e_{SQ} \in E_{SQ}$ and $e_{QT} \in E_{QT}$. By the triangle inequality for $M_\infty$ from Lemma C.4: $w(e_{SQ}) + w(e_{QT}) \geq w(e_{ST}) = W(E_{SQ} \circ E_{QT})$ implies that $W(E_{SQ}) + W(E_{QT}) \geq W(E_{SQ} \circ E_{QT})$ because both terms on the left-hand side are maximized for all edges (not only $e_{SQ}, e_{QT}$) from $E_{SQ}, E_{QT}$. $\square$

We split the proof of Theorem 4.4 into the four shorter parts below:

(a) the completeness and time complexity of the Simpexwise Centered Distribution $\mathrm{SCD}(G)$;

(b) the time complexity of the Simplexwise Bottleneck Metric (SBM) on SCDs;

(c) the time complexity of the Earth Mover's Distance (EMD) on SCDs;

(d) the Lipschitz continuity of SCD in SBM and EMD.

*Proof of Theorem 4.4(a).* As usual, we assume both vertex sets $V(G)$ and $V(G')$ have their centers of mass at the origin $0 \in \mathbb{R}^n$. The proof of invariance and completeness consists of the two parts.

Part *only if* $\Rightarrow$ (invariance). Any orientation-preserving rotation $f$ of $\mathbb{R}^n$ around the origin $0$ bijectively maps the Euclidean graph $G$ to $G'$ and any subset $A \subseteq V(G)$ to $A' = f(A) \subseteq V(G') = f(V(G))$. Since all distances and signs of determinants are preserved, $f$ induces a bijection $\mathrm{OCD}(G) \to \mathrm{OCD}(G')$. Similarly, any orientation-reversing orthogonal map $G \to G'$ indices a bijection $\mathrm{OSD}(G) \to \overline{\mathrm{OSD}}(G') = \mathrm{OSD}(\bar{G}')$.

Part *if* $\Leftarrow$ (completeness). Any bijection $\mathrm{OCD}(G) \to \mathrm{OCD}(G')$ matches $\mathrm{OCD}(G; A \cup \{0\})$ with $\mathrm{OCD}(G'; A' \cup \{0\})$ for some subsets $A \subset V(G)$ and $A' \subset V(G')$ of $n - 1$ vertices. By Lemma C.2(c) any equality $\mathrm{OCD}(C; A) = \mathrm{OCD}(C'; A')$ for $n$-vertex subsets $A, A'$ with aff $= n - 1$ guarantees that $G, G'$ are related by rigid motion in $\mathbb{R}^n$. In the degenerate case, when all subsets have aff $< n - 1$, hence the vertex set $V(G), V(G)'$ belong to a $k$-dimensional subspace of $\mathbb{R}^n$ for $k < n$, we apply the reconstruction of Lemma C.2 to $\mathbb{R}^k$ instead of $\mathbb{R}^n$. In the case $\mathrm{OSD}(C) = \overline{\mathrm{OSD}}(C')$, we get an equality $\mathrm{OCD}(G; A) = \mathrm{OCD}(\bar{G}'; A')$, where $\bar{G}'$ is a mirror image of $G'$. Hence $G, G'$ are related by an orientation-reversing orthogonal map in $\mathbb{R}^n$.

To compute $\mathrm{OCD}(C)$, we consider $\binom{m}{n-1} = \frac{m!}{(n-1)!(m-n+1)!}$ unordered subsets $A \subset V(G)$ of $n - 1$ vertices whose order is arbitrarily chosen. For each fixed $A$, the matrix $D(A \cup \{0\})$ of $\frac{n(n-1)}{2}$ pairwise distances needs $O(n^2)$ time. The Oriented Centered Distribution $\mathrm{OCD}(G; A) = [D(A \cup \{0\}); M(C; A \cup \{0\})]$ includes $n(m - n + 1)$ distances, and also $m - n$ signs and strengths that each requires determinant computations in time $O(n^3)$ by Gaussian elimination. So $\mathrm{OCD}(G; A)$ can be

computed in time $O(n^3m)$. Multiplying the last time by the number $\binom{m}{n-1} = \dfrac{m!}{(n-1)!(m-n+1)!}$ of $(n-1)$-vertex subsets $A \subset V(G)$, the final time for the complete invariant $\text{SCD}(G)$ is

$$O\left(\frac{m!}{(n-1)!(m-n+1)!}n^3m\right) = O\left(m^2(m-1)\dots(m-n+2)\frac{n^3}{(n-1)!}\right) = O\left(\frac{m^n}{(n-4)!}\right)$$

as required. $\qquad\square$

*Proof of Theorem 4.4(b).* The complete bipartite graph $\Gamma(G, G')$ in Definition C.5 has $V = 2\binom{m}{n-1} = O\left(\dfrac{m^{n-1}}{(n-1)!}\right)$ vertices and $E = O\left(\dfrac{m^{2(n-1)}}{((n-1)!)^2}\right)$ edges. The weight $w(e)$ of each edge $e$ equal the metric $M_\infty$, which needs time $O((n^2 + m^{1.5}\log^n m)h!)$ by Lemma C.4. Since $n$ is much smaller than $m$, we can assume that $n^2 \leq O(m^{1.5}\log^n m)$ and drop $n^2$ in further complexities. So we computed the full graph $\Gamma(G, G')$ in time $O\left(\dfrac{m^{2n-0.5}}{(n-1)!}\log^n m\right)$. After that, the bottleneck match $\text{BM}(\Gamma(G, G'))$ can be computed by Hopcroft & Karp (1973) in time $O(E\sqrt{V}) = O\left(\dfrac{m^{2.5(n-1)}}{((n-1)!)^{2.5}}\right)$. The total time for $\text{SBM}(\text{SCD}(G), \text{SCD}(G')) = \text{BM}(\Gamma(G, G'))$ is $O\left(\dfrac{m^{2n-0.5}}{(n-1)!}\log^n m\right) + O\left(\dfrac{m^{2.5(n-1)}}{((n-1)!)^{2.5}}\right) = O\left(\dfrac{m^{2.5(n-1)} + m^{2n-0.5}\log^n m}{(n-1)!}\right).$ $\quad\square$

*Proof of Theorem 4.4(c).* The metric axioms for the Earth Mover's Distance (EMD) are proved in the appendix of Rubner et al. (2000) assuming the metric axioms for the underlying distance $d$, which is the metric $M_\infty$ from Lemma C.4 in our case. The time complexity for EMD follows from the time $O(n!(n^2 + m^{1.5}\log^n m))$ for $M_\infty$ in Lemma C.4, after multiplying by a quadratic factor for the size of cost matrices and adding near-cubic time for the exact computation of EMD, see Fredman & Tarjan (1987); Goldberg & Tarjan (1987). $\qquad\square$

The Lipschitz continuity of SDD in Theorem 3.4(d) needs Lemma B.6 for simpler distributions.

**Lemma C.7** (Lipschitz continuity of $\text{OCD}(G; A)$). *Let $A$ be an ordered sequence of $n-1$ vertices in a graph $G$ in $\mathbb{R}^n$. Let $A'$ and $G'$ be obtained from $A$ and $G$, respectively, by perturbing every vertex in its $\varepsilon$-neighborhood. Then the Oriented Centered Distribution changes in the metric $M_\infty$ from Lemma C.4 by at most $\lambda\varepsilon$, so $M_\infty(\text{OCD}(G; A), \text{OCD}(G'; A')) \leq \lambda\varepsilon$, where $\lambda$ is the Lipschitz constant of the $G$-based distance $d_G$ on vertices of $G$.*

*Proof.* Order all vertices of the given graphs $G, G'$ so that every vertex $p_i \in V(G)$ has the same index as its perturbation $p'_i \in V(G')$. In $\mathbb{R}^n$, the $G$-based distance $d_G(p_i, p_j)$ between any vertices in $V(G)$ changes under perturbation by at most $\lambda\varepsilon$ so that $|d_G(p_i, p_j) - d_G(p'_i, p'_j)| \leq \lambda\varepsilon$.

The 1-1 correspondence $p_i \leftrightarrow p'_i$ induces the (trivial) permutation $\xi$ on $h$ vertices and another (trivial) permutation on $m - h$ columns of the matrix $M(G; A)$. For these fixed permutations, both distances $L_\infty(\xi(D(A)), D(A'))$ and $W_\infty(\xi(M(G; A)), M(G', A'))$ in Lemma C.4 have the upper bound $\lambda\varepsilon$ because all corresponding elements in the underlying matrices change by at most $\lambda\varepsilon$. Taking the minimum for all permutations in Lemma C.4 gives the required upper bound of $\lambda\varepsilon$ for $M_\infty$. $\qquad\square$

*Proof of Theorem 4.4(d).* The upper bound of Lemma C.7 extends to SBM in Definition B.3 and EMD in Definition B.5 because all distances change by at $\lambda\varepsilon$ and the total weight of all flows is 1, so $\text{SBM}(\text{SCD}(G), \text{SDD}(G')) \leq \lambda\varepsilon$ and $\text{EMD}(\text{SCD}(G), \text{SDD}(G')) \leq 2\varepsilon$.

Theorem C.3 was essential to justify a Lipschitz constant $c_n$ of the strength $\sigma(A)$ so that the last coordinates $\frac{s}{c_n}\sigma(A \cup \{0\})$ change by at most $2\varepsilon$ when the bottleneck distance $W_\infty$ is computed on columns in Lemma C.4. $\qquad\square$

