# OpenReview forum: "Complete and continuous representations of Euclidean graphs"
_ICLR.cc/2024/Conference — ICLR 2024 Conference Withdrawn Submission_

### Official Review · Reviewer_C7Vu · 2023-10-28

**Soundness:** 3 good
**Presentation:** 3 good
**Contribution:** 3 good
**Rating:** 3
**Confidence:** 2

**Summary:**

The paper describes descriptors for geometric Euclidean graphs that are invariant under isometry and rigid motion. These descriptors are robust against perturbations in the coordinates of the nodes. Furthermore, the descriptors that are invariant to rigid motion are also complete, meaning these descriptors are injective, and are sufficient to reconstruct the Euclidean graph up to isometry. In addition, the authors show the benefits of their descriptors for the study of molecular graphs.

**Strengths:**

The problem addresses a highly relevant problem within its field. The authors introduce a unique set of descriptors that offer theoretical guarantees in characterizing Euclidean graphs. This contribution underscores the paper's strong theoretical foundation, highlighting its significance in the field. Moreover, the authors do an excellent work in presenting and motivating the problem.

**Weaknesses:**

One notable weakness of the paper lies in the brief and challenging-to-understand experiments section. The lack of clarity in this section makes it difficult to discern the intended purpose of the figures presented. A more detailed explanation or elaboration of the experimental results would greatly enhance the overall comprehensibility of this section.

In addition, the descriptors do not scale effectively with the dimensionality of the Euclidean space and do not consider node or edge attributes, potentially overlooking crucial information in real-world applications where such attributes play a significant role. Moreover, these descriptors can only be utilized by models incorporating some sort of permutation invariance, as they require a "subrepresentation" for each possible permutation of the nodes.

**Questions:**

- In Section 5, the purpose of each figure is unclear. Could you please elaborate on what each figure represents? Additionally, in Table 2, what does SPF stand for?
- Lemma C.2 explicitly states that given the OCD descriptor, the Euclidean graph can be reconstructed up to isometry. However, I could not find a similar result for the SDD descriptor. Does this mean the same principle does not apply to SDD? If that's the case, could you provide the reason for this difference?
- Analogously, it is conjectured that the SDD descriptor  achieves completeness for $h>n-1$, in contrast to the OCD descriptor, which has been proven. What specific factors contribute to the absence of a proof for the SDD descriptor?

---

> ### Author Response · Authors · 2023-11-10
> **Thank you for your helpful comments (part 1)**
>
> Comment. One notable weakness of the paper lies in the brief and challenging-to-understand experiments section. A more detailed explanation or elaboration of the experimental results would greatly enhance the overall comprehensibility of this section.
>
> Answer. We would be happy to answer any specific questions. Since the proposed invariants are the first to be proved complete and Lipschitz continuous for Euclidean graphs, all experiments are completely new, not incremental improvements over any past work.
>
> Briefly, the first part (Table 1) reports hundreds of near-duplicates in QM9 that were never reported because there was no continuous metric. It is important to filter out these near-duplicates for any experiments on QM9, especially those that have too different properties. Indeed, it is physically impossible to change a molecule by a tiny amount and observe a large change in property. At least it's important to know all such cases to continuously quantify the Structure-Property Relationships, see new Problem 2.1.
>
> Second, Figure 6 and Table 2 report the first ever Lipschitz constants for the known properties relative to the first metric (SBM on SCDs) between rigid graphs and to the distances (L_inf and Earth Mover's Distance) on the simpler (incomplete) invariants SDV and PDD.
>
> Question. In addition, the descriptors do not scale effectively with the dimensionality of the Euclidean space
>
> Answer. Quote from page 1: "The most practical cases are the low dimensions n = 2, 3, 4, while the number m of vertices can be much larger. Our main application is for molecular graphs in R^3".
>
> You are right that it would be great to also have a polynomial dependence on dimension. However, there were *no metrics at all* that were computable in polynomial time in the number m of points even for a fixed dimension (even in the plane for n=2).
>
> Question. Do not consider node or edge attributes, potentially overlooking crucial information in real-world applications where such attributes play a significant role.
>
> Answer. Quote right under Figure 5: "Any metrics on vertex attributes and weights can be incorporated into metrics on the invariants below. To make the key concepts clearer, the main paper introduces all invariants for unordered vertices without attributes and only for sign weight".
>
> More importantly, the first ever duplicate analysis of QM9 in Table 1 shows that all molecules can be distinguished only by their vertex positions and presence/absence of bonds even without any atomic types.
>
> This experiment reflects our basic physical intuition because we cannot replace one atom by different one without changing geometry. However, this principle was impossible to verify without complete invariants with continuous metrics.
>
> Comment. these descriptors can only be utilized by models incorporating some sort of permutation invariance, as they require a "subrepresentation" for each possible permutation of the nodes.
>
> Answer. A simple way to guarantee permutation invariance is to convert any unordered set of numbers a_1,...,a_m into their moments: the average (a_1+...+a_m)/m, the standard deviation, and so on. Any set can be reconstructed (up to permutation) from such symmetric polynomials from m moments, see https://en.wikipedia.org/wiki/Symmetric_polynomial
>
> Question. in Table 2, what does SPF stand for?
>
> Answer. SPF stands for Structure-Property Factor. This is the Lipschitz constant in front the of the metric needed to quantify a difference in property, see the caption of Table 2.
>
> Question. Lemma C.2 explicitly states that given the OCD descriptor, the Euclidean graph can be reconstructed up to isometry. However, I could not find a similar result for the SDD descriptor.
>
> Answer. Yes, the Simplexwise Distance Distribution (SDD) is for any metric (not necessarily Euclidean) space. The reconstruction is proved for invariants (OCD, SCD) in section 4 for Euclidean space. The letter C refers to the center of mass, which is defined by the linear combination of given points, hence makes sense only in a vector space.
>
> Question. Does this mean the same principle does not apply to SDD? If that's the case, could you provide the reason for this difference? Analogously, it is conjectured that the SDD descriptor achieves completeness for in contrast to the OCD descriptor, which has been proven. What specific factors contribute to the absence of a proof for the SDD descriptor?
>
> Answer. Yes, you are right that the Euclidean structure was essentially used for completeness and reconstruction. For example, the position of any point in R^3 is determined by 3 distance to other 3 fixed points (not in the same line) and a sign. In an arbitrary non-Euclidean space, there is no similar rigidity of positions with respect to distances. We conjecture that SDD of a high enough order is complete in any metric space, so this problem is left for future research.
>
> We will reply to the remaining question in the second part below.

---

> > ### Author Response · Authors · 2023-11-10
> > **Answer to the remaining question about experiments in section 5**
> >
> > Questions. Could you please elaborate on what each figure represents?
> >
> > Figure 6 has three plots with the property of free energy (Q) on the vertical axis. This energy is the most important property characterizing the thermodynamic stability (real existence) of a molecule.
> >
> > On the horizontal axis, each plot has one of three distances:
> >
> > L_infinity metric between Sorted Distance Vectors SDVs (easy and incomplete invariant),
> >
> > Earth Movers Distance (EMD) between Pointwise Distance Distribution PDDs (more sophisticated but still incomplete invariant in R^3), and
> >
> > the new Simplexwise Bottleneck Metric SBM between Simplexwise Centered Distributions SCDs (new complete and Lipschitz continuous invariant).
> >
> > The first plot for the easy invariants SDV shows that the energy (relatively) rapidly changes with respect to L_infinity because this distances is not a metrics (doesn't distinguish all possible Euclidean graphs).
> >
> > The second and third plot show that the energy changes slower with respect to EMD (almost a metric, though there are know counter-examples in R^3, though not in QM9) and the proper metric SBM.
> >
> > The numerical parameters of these Structure-Property Relationships: factor SPF and deviation SPD are reported in Table 2.
> >
> > Figure 7 contains three projections of 130K+ molecules of QM9 to the coordinates x,y, expressed via the minimum, median, and maximum of inter-atomic distance. The color in the first plot is the free energy (G).
> >
> > The color in second plot shows the energy barrier of every molecule, which was not possible to calculate before without a proper metric. We simply didn't know with 100% certainty how close molecules were to each other because complete invariants with a continuous metric were needed.
> >
> > The second plot highlights that only a small proportion of QM9 molecules have high energy barriers (sit in deep local minima), all others are in shallow local minima. This is the first view of the new geographic-style landscape of QM9. This deep-vs-shallow local minima analysis is now possible for any graph-based data with properties due to Theorem 4.4.
> >
> > The third plot shows the average slope of every QM9 molecular relative to its 5 neighbors. Using again a geographic analogy, most molecules sit roughly at the same altitude in their neighborhoods, but a few molecules lie on more inclined slopes and can be unstable not because of prohibitively high energy values but due to the surrounding energy landscape, which was previously impossible to describe without complete invariants and continuous metric.
> >
> > As promised in "This section describes three advances that were impossible to achieve by any past tools", Theorem 4.4 enabled breakthroughs far beyond incremental improvements.
> >
> > We are happy to answer more questions. Thank you.

---

> > > ### Comment · Reviewer_C7Vu · 2023-11-16
> > >
> > > Thank you for the explanation. Some points are clearer now, however I still have some doubts regarding the figures.
> > >
> > > Regarding Figure 6: If the horizontal axis represents the metric value between two points, how do I interpret the energy of a pair of points. Is the difference of the energy values? What does exactly represent the green line in Figure 6? You mention that in the first plot the energy rapidly changes with respect to the the L_infinity distance, while for the other metrics this is not the case. However, I can not identify such patterns from the scatter plots. Should the green lines aid to detect the rapid change? What I can detect, is that for the first plot, more points concentrate in the bottom right corner of the plot, meaning that there pairs of points that are farther to each other with respect to the L1_infinite distance, yet with similar free-energy (asuming the energy of a pair of point is the difference of their energies) . This pattern is not as prominent for the other metrics. Why should this be related with a rapid change? I find difficult to interprete correctly these plots, as I have the impression that I am misunderstanding something.
> > >
> > >
> > > Regarding table 2: Is the SPF value equivalent to the slope of the green line in Figure 6? Analogously is SPD the distance at which the slope suddenly changes? How should be this change of slope interpreted? Also, why is there such a discrepancy for the values of the energy U0? The  SPF and SPD values of the L_infinity and SBM columns are similar, but the ones of the EMD column are totally different. If my understanding is correct, it seems as the EMD is the best metric since their slopes are lower? Is that so?
> > >
> > > Regarding the permutation invariance: I do not think the permutation invariance can be solved by the computation of the moments. The SDD and SCD descriptors are formed by sets of pairs of matrices. Each pair of matrices is determined by a h-vertex subset A of V(G). Depending on the order of the h-vertex subset, the associated pait of matrix will also change the order of its entries. Thus, in order to process efficiently each pair of matrices, one needs to be permutation invariant.

---

### Official Review · Reviewer_AGV3 · 2023-10-28

**Soundness:** 3 good
**Presentation:** 4 excellent
**Contribution:** 2 fair
**Rating:** 8
**Confidence:** 3

**Summary:**

This paper gives a new signature suitable for representing geometric graphs that's invariant under isomorphism (permutations of vertices). It first gives desired properties of such invariants, in particular, stability under vertex perturbations, and then gives one based on distances after suitable shifts. The efficiency of the algorithm is analyzed, and experiments were done on finding (near) duplicates in the QM9 molecules dataset.

**Strengths:**

The algorithm given is natural, and has a high degree of interpretability. The experiments were performed on real data, and give direct comparisons with previous works (using significantly different invariants).

**Weaknesses:**

The theoretical running times of the algorithms still have exponential dependences (albeit on a different parameters). The requirements given in Problem 1.1, while natural, is also quite complex. These are understandable to me due to the complex domain-specific info of the molecules though.

**Questions:**

My background is in combinatorial/algebraic graph algorithms, so am accustomed to features based on eigenvalues. Are there graphs with same/similar list of eigenvalues where the invariants introduced here differ?

---

> ### Author Response · Authors · 2023-11-11
> **Thank you for your helpful support**
>
> Comment. The theoretical running times of the algorithms still have exponential dependences (albeit on a different parameters).
>
> Answer. The key (large) parameter is the number m of points. The practical dimensions are n=2,3. The order parameter h, which equals n-1 in the Euclidean case, can be also chosen small, because already for h=1 the resulting invariant PDD is generically complete.
>
> Comment. The requirements given in Problem 1.1, while natural, is also quite complex. These are understandable to me due to the complex domain-specific info of the molecules though.
>
> Answer. Several paragraphs after Problem 1.1 discussed all the conditions as a basic minimum for any data objects (instead of graphs) and equivalence (instead of isometry). So Problem 1.1 provides a well-justified target for data science far beyond molecules.
>
> Question. My background is in combinatorial/algebraic graph algorithms, so am accustomed to features based on eigenvalues. Are there graphs with same/similar list of eigenvalues where the invariants introduced here differ?
>
> Answer. Yes, three eigenvalues of any point cloud in Euclidean 3-dimensional space are easy isometry invariants. These three numbers are very far from being complete descriptors of arbitrary point clouds. However, next week we can produce analogs of Figure 7 by using three eigenvalues and also list pairs of molecules with almost identical eigenvalues.
>
> We are happy to answer more questions. Thank you.

---

### Official Review · Reviewer_ekKc · 2023-10-29

**Soundness:** 3 good
**Presentation:** 3 good
**Contribution:** 2 fair
**Rating:** 5
**Confidence:** 5

**Summary:**

The paper defines Euclidean graphs as collections of m points in R^n, some of which are connected by straight non-intersecting lines (edges). The paper searches and finds descriptors for such graphs which satisfy several desirable axioms: invariance to rigid motions and permutations, completeness, LIpschitzness with respect to some known metric.

The authors provide experiments which show the QM9 dataset contains some multiple molecules (and some other experiments which were not very well explained)

**Strengths:**

There have been several papers on similar topics in the last 1-2 years. This paper seems to be the first to
(i) show completeness in a setting where edges are given. Many previous works only focused on the setting where there is no graph structures.
(ii) Show that the invariants described are Lipschitz.

**Weaknesses:**

(a) The proof of the first contribution is very simple and I'm not sure it is correct, see questions below.
(b) While previous papers have not considered Lipschitzness, this is probably not so difficult to achieve with other methods. The more interesting challenge IMHO lies in being bi-Lipschtz with respect to an appropriate invariant metric on the input space. For a definition of bi-Lipschtizness and a technique which is helpful for obtaining Lipschitz invariants from non-Lipschitz invariants See e.g.
[COMPLETE SET OF TRANSLATION INVARIANT MEASUREMENTS WITH LIPSCHITZ BOUNDS by Cahill et al.]
(c) The experiments were not well explained. They do not seem to point to an advantage of the described descriptor over other descriptors discusses in the literature.
(d) the paper is overall well written but overemphasizes things which were already discussed in the literature (invariance, completeness) and doesn't talk enough about the unique contributions of the paper: The metric used is not defined in the paper but only in the appendix. The experiments are not explained as stated above. No intuition regarding proofs is provided- and in particular in the technical novelty with respect to earlier work.

**Questions:**

Regarding the proof of completeness in Lemma C.2 part (c). You say
"the presence of an edge... is determined... by the matrices D(A \cup 0) and M(G;A \cup 0)"
I'm not sure about this argument. Say you have two points p,p' which are both not in A. How would you know if there is an edge between them from the matrices D(A \cup 0) and M(G;A \cup 0)?

Other questions:
* Where in your proof do you need straight non-intersecting edges?
* Also is this assumption always valid in chemistry? If so could you explain/ give reference?
* Can you explain in more detail what problem 2.1 is? Is this a problem you solve in this paper?
* Page 5: what is the `main paper'? There is only one paper. How are these definitions related to the definitions above when you defined
  IGS and RGS?
* A paper you could consider adding to your refs:
[Is Distance Matrix Enough for Geometric Deep Learning? Li et al]


Local remarks and suggestions regarding writing (no need to address in rebuttal):
* Last paragraph in page 2 is confusing I would rephrase
* In the past work section: Why do you say Problem 1.1 is much simpler? The explanation gives is that there is an  algorithm with exponential complexity to solve the problem (note also this algorithm has issues when there are eigenvalue multipliciities)
* There are two  things which seem to me make the presentation unnecessarily too complicated (this is not essential). Firstly, why do you decide to have a (n-1) by (n-1) distance matrix instead of n by n like the rest of the world uses? It makes the notation cumbersume. Secondly, why do you bother to lexicographically sort things when later you will define a permutation invariant distance? You can just store things in an arbitrary order.
* I think the first paragraph is Section 5 is unnecessarily aggressive. Firstly, I didn't attend Isabelle Guyon's talk but I would guess the remark you are quoting was made half-jestingly. Secondly, experiments on QM9 use the standard train/test paradigm and results are evaluated on the test. This is a reasonable and standard practice. The limitations of this practice regarding generalizing to out of distribution data are also known. Since you are not addressing these issues in the experiments I don't see why you need to bring up the whole issue. Just say what your experiments are.
* Page 12 just before the beginning of the appendix there is a paragraph which maybe is misplaced?

---

> ### Author Response · Authors · 2023-11-10
> **Thank you for the detailed review and helpful questions.**
>
> Question. Regarding the proof of completeness in Lemma C.2 part (c). Say you have two points p,p' which are both not in A. How would you know if there is an edge between them from the matrices D(A \cup 0) and M(G;A \cup 0)?
>
> Answer. The last two lines in the proof of Lemma C.2(c) say that "The presence (or absence) of an edge between any p, q in V (G) is determined by the sign +1 (or −1, respectively) of each distance dG(p,q) in the matrices D(A union {0}) and M(G; A union {0})".
>
> By Definition 3.3 all distance in the matrices D and M are *G-based* and denoted by d_G(p,q). This notation was introduced before Definition 3.1:
>
> "To make the key concepts clearer, the main paper introduces all invariants for unordered vertices without attributes and only for sign weights: w[p, q] = +1 if the vertices p, q are connected by an edge of G, otherwise w[p, q] = −1. Let dG(p, q) denote any G-based distance between p, q in V (G). We can take the signed distance w[p, q]d(p, q)".
>
> So, any distance in all matrices has a sign: +1 means an edge, -1 means no edge.
>
> Question. Where in your proof do you need straight non-intersecting edges?
>
> Answer. You are right that the condition on non-intersecting edges is superfluous. We need only a combinatorial edge given by a pair of points. The condition of a straight edge is needed only to draw a straight line. Without this condition, the reconstruction produces positions of all vertices (up to rigid motion) and indicators which vertices should be connected.
>
> Question. Also is this assumption always valid in chemistry? If so could you explain / give reference?
>
> Answer. Yes, molecules are often represented by atoms connected by straight chemical bonds. We can add the reference to the book "Chemical graph theory: introduction and fundamentals" by D Bonchev, 1991. Our edges are represented by abstract yes/no links.
>
> Question. Can you explain in more detail what problem 2.1 is? Is this a problem you solve in this paper?
>
> Answer. Problem 2.1 is the first formulation (to the best of our knowledge) of the continuous version of the Structure-Property Hypothesis (SPR). The original SPR informally says that "a structure determines all properties". In the mathematical language, if two structures are rigidly equivalent, they should have same properties. In this exact formulation, the last statement is rather trivial.
>
> However, any real structures cannot be exactly equivalent, they always differ by some noise in measurements. Hence the state the continuous (practical) version of the SPR hypothesis as Problem 2.1: briefly, how close should be two structures (in terms of a metric between rigid classes) to guarantee that they have similar properties? More exactly, determine the Lipschitz constant of a property relative to perturbations of a structure.
>
> Since there was no metric on rigid classes of Euclidean graphs, Problem 2.1 made no sense. Indeed, if the first axiom fails, we have two non-equivalent graphs G,H with potentially different properties but d(G,H)=0, so the property is discontinuous with respect to this d.
> Indeed, the distance d sees no difference between G,H but their properties can differ!
>
> Now, after complete invariants with Lipschitz continuous metrics are finally found, Problem 2.1 opens doors to rigorous investigations of the property landscapes on a continuous and complete invariant-based map, which was started in section 5.
>
> Question. What is the `main paper'? There is only one paper. How are these definitions related to the definitions above when you defined IGS and RGS?
>
> Answer. The words `main paper' referred to the 9-page main part before the references. Definition 3.1 reminds about the simpler (incomplete) invariants for the ablation study in section 5. The spaces RGS and IGS require complete invariants (SCD) in Definition 4.1.
>
> Comment. A paper you could consider adding to your refs: [Is Distance Matrix Enough for Geometric Deep Learning? Li et al]
>
> Answer. Thank you for your helpful reference, which we will certainly add. The distance matrix can be used when all given points (or vertices in a graph have distinct labels or indices). This case of ordered points was also discussed in paragraph 2 on page 4.
>
> Question. In the past work section: Why do you say Problem 1.1 is much simpler?
>
> Answer. The full quote is "Problem 1.1 is much simpler for a cloud C of m unordered points when a graph has only isolated vertices and no edges."
>
> Indeed, there are m(m-1)/2 potential edges between m points. Hence, for a fixed set of vertices in R^n, there are exponentially many non-isometric graphs on these fixed m vertices (even if we consider potential symmetries), so edges add a lot of complexity to a point cloud.
>
> Further questions are answered in the second part below because of the character limit.

---

> > ### Author Response · Authors · 2023-11-10
> > **Part 2 of answers**
> >
> > Question. (this is not essential). Firstly, why do you decide to have a (n-1) by (n-1) distance matrix instead of n by n like the rest of the world uses?
> >
> > Answer. Sorry, this was some over-optimization. Yes, n-by-n matrices are fine.
> >
> > Question. Secondly, why do you bother to lexicographically sort things when later you will define a permutation invariant distance? You can just store things in an arbitrary order.
> >
> > Answer. You are right. Again, it is probably unnecessary over-optimization.
> >
> > Question. I didn't attend Isabelle Guyon's talk but I would guess the remark you are quoting was made half-jestingly.
> >
> > Answer. Yes, it was an informal call to try harder and work towards complete invariants (meaning no false negatives and no false positives for all possible data, not only on a specific dataset), which was done in this paper for all Euclidean graphs.
> >
> > Question. experiments on QM9 use the standard train/test paradigm and results are evaluated on the test. This is a reasonable and standard practice. The limitations of this practice regarding generalizing to out of distribution data are also known. Since you are not addressing these issues in the experiments I don't see why you need to bring up the whole issue.
> >
> > Answer. The experiments in section 5 address the past limitations of descriptors or distances that allowed zero values for non-equivalent structures. The first experiment in section 5 reports hundreds of near-duplicates in Table 1 (actually, exact duplicates if we take into account floating point errors) in QM9, which were nether reported because there was no metric on rigid classes of Euclidean graphs.
> >
> > Some of these near-duplicates have different properties as shown in the zoomed-in images of Figure 6. This analysis of the continuous SPR hypothesis was impossible by descriptors with false positives meaning non-equivalent graphs with the same value of the descriptor.
> >
> > Question. Page 12 just before the beginning of the appendix there is a paragraph which maybe is misplaced?
> >
> > Answer. You are right that this introduction to the appendices could have been better highlighted.
> >
> > We would be happy to answer any more questions.

---

> > ### Comment · Reviewer_ekKc · 2023-11-12
> > **Still not convinced proof is correct**
> >
> > Regarding the question about the proof of Lemma C.2: I understand that if you know the signed distance between p and q then this tells you if there is an edge. The issue is that you don't have the labelled distance matrix. You have the pairwise distances between points in A, and the pairwise distances between the points in A and the other points. So if p and q are both not in A, you will not know d(p,q) and so will not know the sign, and so will not know if there is an edge.
> >
> > Thus I still think the proof of this lemma is incorrect.

---

> > > ### Author Response · Authors · 2023-11-12
> > > **thank you for your helpful clarification**
> > >
> > > Comment. if p and q are both not in A, you will not know d(p,q) and so will not know the sign, and so will not know if there is an edge.
> > >
> > > Answer. Yes, you are right that the Oriented Centered Distribution OCD(G;A) allows us to reconstruct the vertex set V(G) of a graph G uniquely up to rigid motion and all identify the presence/absence of edges only from vertices of A but not between vertices from V(G)-A. Let as mark all connections from p in A to q in V(G)-A by a solid edge (if the distance sign is +1) or a dashed edge (if the distance sign is -1). All other connections between points of V(G)-A are uncertain at the moment. We agree that OCD(G;A) in Lemma C.2(c) should be replaced with the full invariant SCD(G) by using the following extra argument.
> > >
> > > Let us take another subset A' of n-1 vertices. The previous argument again uniquely reconstructs the vertex set V(G). Now we can mark more edges (from points of A') as solid (present) or dashed (absent). Since the reconstruction of V(G) is unique up to rigid motion, though we don't know vertex correspondences, the two reconstructions of V(G) can be uniquely matched by rigid motion. Hence we will know more edges (present or absent) from the union of A and A'. We continue using other (n-1)-vertex subsets of V(G) until all edges are determined. The extra reconstructions increase the complexity but the worst upper bound for the number of subsets A is "m choose n-1", hence polynomial in m as required. Do you agree with this extended proof?
> > >
> > > Thank you again for your important comment.

---

> > > > ### Comment · Reviewer_ekKc · 2023-11-13
> > > > **Still not convinced**
> > > >
> > > > Thanks for the answer.
> > > >
> > > > You say "two reconstructions of V(G) can be uniquely matched by rigid motions" This is not true iMHO if the point cloud has symmetries, e.g., if the point cloud is the corners of a cube. Or some more complicated "something-hedron".

---

> > > > > ### Author Response · Authors · 2023-11-13
> > > > > **Thank you for mentioning the symmetries**
> > > > >
> > > > > Comment. You say "two reconstructions of V(G) can be uniquely matched by rigid motions" This is not true iMHO if the point cloud has symmetries.
> > > > >
> > > > > Answer. V(G) denotes only the set of vertices without edges. You probably meant that there could be a non-trivial isometry that maps the vertex set V(G) to itself (as a cloud of unordered points), which may not respect all edges of the Euclidean graph G.
> > > > >
> > > > > It can be easier to clarify first the dimension n=2. Then SCD(G) of a graph G with m vertices in the plane is the unordered collection of m OCDs (one per vertex). Each of these OCDs allows us to reconstruct a vertex-star S(v) that is a geometric graph on the full vertex set V(G) and edges (present or absent) only from one vertex v to all other vertices from V(G). You are right that this set of vertex-stars (for h=n-1=1) may indeed be insufficient to uniquely reconstruct the whole graph G in symmetric cases.
> > > > >
> > > > > Hence we should use the stronger invariant SCD for h=2, which contains m(m-1)/2 OCDs for pairs of vertices. Then each new OCD (based on two vertices v_1,v_2 instead of one) allows us to reconstruct an edge-star S(v_1,v_2) that is a graph on V(G) with more known edges (present or absent) from v_1 and v_2 to all other vertices from V(G).
> > > > >
> > > > > Each of the simpler m vertex-stars should be shared by at least m-1 larger edge-stars because the invariant is realized by some graph G. If one (any) vertex-star is shared by exactly m-1 edge-stars, the whole graph G is reconstructed from these uniquely determined m-1 edge-stars, which now contain enough information about the presence or absence of all edges of G.
> > > > >
> > > > > If the vertex set V(G) is highly symmetric, several vertex-stars can be equivalent, which can be checked by EMD in the same sense as their algebraic versions OCD for h=1 in Definition 4.1. Then every such repeated vertex-star should be shared by a multiple k(m-1) of edge-stars because each all vertex-stars are in a bijection with all vertices from V(G).
> > > > >
> > > > > Since SCD is geometrically realized by some G, these k(m-1) edge-stars should split into k subsets, where each subset of m-1 edge-stars gives rise to the (unique up to rigid motion) graph G. This splitting is unique (up to swaps of equivalent vertex-stars) because only a correct combination of vertex-stars gives rise to the geometrically unique vertex set V(G) up to rigid motion.
> > > > >
> > > > > These k subsets (of m-1 edge-stars in each) are equal to each other because each geometrically realizable set of m-1 edge-stars uniquely determines the whole graph G with all edges. An equality between these k subsets may come from (say) a rotational symmetry in the plane.
> > > > >
> > > > > The extended argument above essentially used the stronger invariant SCD with h=2. However, the ambient dimension n was not important because m-1 edge-stars sharing a common vertex-star uniquely give rise to the whole graph G with all edges in any dimension n>=2.
> > > > >
> > > > > In conclusion, we agree that in the plane case n=2, the SCD invariant for h=1 was almost complete but unfortunately insufficient in singular cases, which actually shows that the case of graphs is much harder than point clouds. However, the choice h=n-1 still works for all higher dimensions n>=2 because edge-stars suffice to recover the presence/absence of all edges of G.
> > > > >
> > > > > Thank you again for your helpful comment. May we ask if you are convinced now?

---

> > > > > > ### Comment · Reviewer_ekKc · 2023-11-14
> > > > > > **Still don’t agree**
> > > > > >
> > > > > > “Each of the simpler m vertex-stars should be shared by at least m-1 larger edge-stars because the invariant is realized by some graph G. If one (any) vertex-star is shared by exactly m-1 edge-stars, the whole graph G is reconstructed from these uniquely determined m-1 edge-stars, which now contain enough information about the presence or absence of all edges of G.“
> > > > > >
> > > > > > I don’t agree: construct a point cloud in the plane by taking uniformly spaced points on the unit circle. Build a graph on these points. Our goal is to reconstruct this graph. Now add a point in the origin and connect it to all other points.
> > > > > >
> > > > > > The origin’s vertex star is unique so it will be shared by exactly m-1 edge stars. I don’t see how this helps in reconstructing the graph.

---

> > > > > > > ### Comment · Reviewer_ekKc · 2023-11-14
> > > > > > > **Setting boundaries**
> > > > > > >
> > > > > > > I don’t think it is my job as a reviewer to repeatedly review imperfect proofs. The proof in the submission was wrong. Please make one final revision and choose one of two options:
> > > > > > > either send me a proof you are *sure is true* and i will judge it one last time. I will lower my score if i believe it is incorrect.
> > > > > > >
> > > > > > > Or
> > > > > > > retract this claim and try to defend the paper without it

---

> > > > > > > > ### Author Response · Authors · 2023-11-17
> > > > > > > >
> > > > > > > > Comment. The origin’s vertex star is unique so it will be shared by exactly m-1 edge stars. I don’t see how this helps in reconstructing the graph.
> > > > > > > >
> > > > > > > > Answer. This rotational symmetric graph with one singular vertex cannot be a counter-example to the completeness of an invariant based on two arbitrary vertices.
> > > > > > > >
> > > > > > > > Comment. I don’t think it is my job as a reviewer to repeatedly review imperfect proofs.
> > > > > > > >
> > > > > > > > Answer. If you have seen a perfect proof of a non-trivial result at ICLR, we would be happy to follow this example.
> > > > > > > >
> > > > > > > > Comment. I will lower my score if i believe it is incorrect.
> > > > > > > >
> > > > > > > > Answer. We never mentioned scores because we care more about science. Since the scientific advances are not appreciated, we will withdraw the paper to avoid wasting time.

---

### Official Review · Reviewer_FYHL · 2023-10-31

**Soundness:** 2 fair
**Presentation:** 3 good
**Contribution:** 2 fair
**Rating:** 3
**Confidence:** 4

**Summary:**

This paper focuses on the construction of descriptors for Euclidean graphs. A Euclidean graph is a point cloud in $\mathbb{R}^n$, some of which are connected by (non-intersecting) straight lines.

The descriptors under consideration should be complete invariants, i.e. they should allow the identification of any graph up to rigid motion. Furthermore, they should be robust to noise in the following sense: if a vertex is moved by $\epsilon$, then the descriptor should be perturbed by $\epsilon$ up to a multiplicative constant, for some metric to be determined. Finally, both the descriptor and the metric should be computable in polynomial time.

The authors propose such an invariant for graphs in metric space then Euclidean graphs, and derive its time complexity.

The entire methodology has been developed for the analysis of molecular graphs whose vertices are atomic centers and edges represent inter-atomic bonds. A numerical application to a database of 130,808 molecules illustrates the behavior of the proposed algorithms and highlights their relevance.

**Strengths:**

The question raised by the paper is an interesting one, and goes beyond the application to molecular graphs. In addition to control its complexity, requiring the descriptor to be Lipschitz is, in my opinion, relevant in many contexts. The solution developed in the paper looks promising and is supported by rigorous theoretical arguments.

The introductory section is highly pedagogical and provides a clear understanding of the issues at stake in the paper.

**Weaknesses:**

When we read the motivations of the paper, we expect the graphs considered to have a large number of vertices, which could exclude some algorithms from the literature. In particular, the authors insist on the need for polynomial complexity in the number $m$ of vertices.

However, if the complexities are indeed polynomial, the method may turn out to be too slow for large graphs with a term in $m^5$ in Theorem 4.4 (in dimension 3).

The application considered in the numerical part actually deals only with small graphs ($m<30$) for which many other methods (even with worse complexities) could be employed.
It seems to me that this limits the interest of the proposed method, both for the example considered in the paper (where other algorithms could be applied) and for application to larger graphs (where the time complexity could be too bad).

In addition, I think it is a good point to control descriptor variations, but limiting them to a single vertex seems restrictive: in real data, all the vertices are subject to noise. The result obtained is interesting but should be generalized to metrics that take into account all the vertices of the graph.

Finally, the paper lacks a numerical section that illustrates the algorithm's behavior on synthetic graphs of varying size, both in terms of computation time and robustness to noise, with comparisons with others methods from the literature, e.g. graph distances or kernels).

**Questions:**

From my point of view, a numerical part with computation times should illustrate the theoretical results of complexity to highlight which size of graph can be handled by the proposed descriptors. Do the authors have an answer to this point?

I do not understand the role of $h$ in section 3 and more precisely in Theorem 3.4. To minimize complexity, we want to take $h=1$, but is the descriptor still relevant (in relation with the remark at the end of page 6)? How $h$ is selected and what happens to complexity in that case?

In the same vein, what is the exact role of $l$ and how should it be selected, both in Theorems 3.4 and 4.4? Is it a way to control the time complexity?

How were these two parameters selected in the numerical section of the paper?

---

> ### Author Response · Authors · 2023-11-10
> **Thank you for the helpful questions**
>
> Comment. When we read the motivations of the paper, we expect the graphs considered to have a large number of vertices, which could exclude some algorithms from the literature.
>
> Answer. The abstract talks about applications to molecular graphs from QM9, which contains 130K+ molecules of up to 29 atoms each.
>
> Comment. If the complexities are indeed polynomial, the method may turn out to be too slow for large graphs with a term in in Theorem 4.4 (in dimension 3).
>
> Answer. The new invariant SCD is the final in the hierarchy of the simpler and faster invariants SDV and PDD. The experiments in section 5 apply these three invariants in the hierarchical way, first rough separation by SDV, then finer distribution by PDD and the final and definite answer about duplication and proper metric by the complete invariant SCD.
>
> Comment. The application considered in the numerical part actually deals only with small graphs for which many other methods (even with worse complexities) could be employed.
>
> Answer. The detailed review of related work in section 2 thoroughly described that all past methods didn't solve full Problem 1.1. If we accidentally missed any specific references, please let us know and we would be happy to include them.
>
> Comment. For the example considered in the paper (where other algorithms could be applied) and for application to larger graphs (where the time complexity could be too bad).
>
> Answer. The completeness and Lipschitz continuity are more important than computability due to the faster invariants SDV and PDD similarly based on inter-atomic distances.
>
> Comment. In addition, I think it is a good point to control descriptor variations, but limiting them to a single vertex seems restrictive: in real data, all the vertices are subject to noise. The result obtained is interesting but should be generalized to metrics that take into account all the vertices of the graph.
>
> Answer. In Theorem 3.4(c) and 4.4(c) the words "perturbing any vertex" meant "perturbing each vertex" (all vertices, not a single one).
>
> Comment.the paper lacks a numerical section that illustrates the algorithm's behavior on synthetic graphs of varying size, both in terms of computation time and robustness to noise, with comparisons with others methods from the literature, e.g. graph distances or kernels).
>
> Answer. The algorithm's behavior with respect to size and robustness to noise were theoretically proved for all graphs. One can ask for synthetic experiments when there are no theoretical proofs. However, we have completed this stage with 100% guarantees and considered only real data such as 130K+ molecules from QM9.
>
> Question. From my point of view, a numerical part with computation times should illustrate the theoretical results of complexity to highlight which size of graph can be handled by the proposed descriptors. Do the authors have an answer to this point?
>
> Answer. Ok, next week we'll run extra experiments on random graphs with a growing number of vertices.
>
> Question. I do not understand the role of h in section 3 and more precisely in Theorem 3.4. To minimize complexity, we want to take h=1, but is the descriptor still relevant (in relation with the remark at the end of page 6)?
>
> Answer. This parameter is more relevant for general metric spaces. Yes, SDD for h=1 is the past invariant PDD, SDD for h=2 is a stronger invariant, SDD for h=3 is even stronger and so on. In Euclidean space, Theorem 4.4 proves that h=n-1 provides a complete invariant in R^n.
>
> Question. How h is selected and what happens to complexity in that case?
>
> Answer. h=1 gives the strong invariant PDD. h=2 gives SDD that distinguished all known counter-examples to the past invariants in R^3. All complexities explicitly include h.
>
> Question. In the same vein, what is the exact role of l and how should it be selected, both in Theorems 3.4 and 4.4? Is it a way to control the time complexity?
>
> Answer. The parameter l is determined in Theorem 3.4(c) as the size (number of non-equivalent sets) and has the explicit upper bound in terms of m and n (or h), see "let SDD(G; h) and SDD(G′; h) have a maximum size l", similarly in Theorem 4.4(c), .
>
> Question. How were these two parameters selected in the numerical section of the paper?
>
> Answer. Molecular graphs are in R^3, so n=3. Then h=n-1=2 for the SCD invariant.
>
> We are happy to answer any more questions.

---

### Official Review · Reviewer_7QMR · 2023-11-06

**Soundness:** 4 excellent
**Presentation:** 3 good
**Contribution:** 3 good
**Rating:** 6
**Confidence:** 2

**Summary:**

The paper delves into the representation and comparison of Euclidean graphs using concepts developed for point clouds. These point clouds are often represented by finite sets of unlabeled points, and the most natural equivalence for these point clouds is rigid motion or isometry, which maintains all inter-point distances.

The paper emphasizes the importance of complete isometry invariants, which are descriptors that can reliably compare Euclidean graphs without ambiguity. The authors propose the first continuous and complete invariant for Euclidean graphs, which is computable in polynomial time based on the number of points.

Furthermore, the paper also touches upon the challenges posed by real data, which is often noisy, and provides implementation details and code for verification.

**Strengths:**

The paper addresses an important problem of checking invariance among Euclidean graphs and presents a pioneering approach to the challenge of recognizing patterns in Euclidean graphs.

One of the standout strengths is the introduction of the first continuous and complete invariant for Euclidean graphs This invariant not only offers a reliable comparison but is also computable in polynomial time based on the number of points, showcasing its potential efficiency.

The authors provide a sufficient discussion on the significance of distinguishing graphs related by isometry or having similar properties.

Additionally, the paper's acknowledgment of real-world challenges, such as noisy data demonstrates a comprehensive understanding of practical scenarios, making the proposed solutions more applicable and relevant.

**Weaknesses:**

Following would be some of the central points of discussion :

1) **Scalability \& Approximation**: While this is a pioneering effort for complete invariance in graphs, to be applicable to a much broader application class the question of scalability is crucial. Graphs typically comprise thousands or millions of nodes. Having a polynomial complexity on the nodes makes it harder to execute. To this end, would the authors think of some approximations that could be possible? or as a reverse question can comment on the impossibility of such a scenario?

2) **Other graph types**: I am assuming the graphs in the paper are undirected and consist of an edge weight of 1. Can the theory developed extend easily to weighted and/or undirected graphs? What would be the challenges in doing so?

3) **Robustness to noise**: This is a minor point, but is there some theoretical result possible on the tolerance to outliers or Gaussian noise levels?

**Questions:**

Please refer to **weaknesses** section

---

> ### Author Response · Authors · 2023-11-11
> **Thank you for your supportive review**
>
> Comment. Scalability & Approximation: While this is a pioneering effort for complete invariance in graphs, to be applicable to a much broader application class the question of scalability is crucial. Graphs typically comprise thousands or millions of nodes.
>
> Answer. The abstract emphasized twice that the key target application is for molecular graphs. The mentioned database QM9 consists of 130K+ molecules that have up to 29 atoms.
>
> Comment. Having a polynomial complexity on the nodes makes it harder to execute. To this end, would the authors think of some approximations that could be possible? or as a reverse question can comment on the impossibility of such a scenario?
>
> Answer. Approximations are always possible. The weaker (but generically complete) invariant PDD (Pointwise Distance Distribution) has a near-linear time in the number k of neighbors.
>
> These k nearest neighbors can be computed by a fast nearest neighbor search in any metric space, see the latest results in Elkin et al "A new near-linear time algorithm for k-nearest neighbor search using a compressed cover tree" (ICML 2023).
>
> Comment. Other graph types: I am assuming the graphs in the paper are undirected and consist of an edge weight of 1. Can the theory developed extend easily to weighted and/or undirected graphs? What would be the challenges in doing so?
>
> Answer. The edges are undirected but lengths of edges can be arbitrary. Section 3 defines strong invariants SDD (Simplexwise Distance Distribution) for finite graphs in any metric space (or weighted undirected graphs in your words). For example, a graph can be built on any finite set of vertices with the shortest path metric for any positive lengths of edges.
>
> Question. Robustness to noise: This is a minor point, but is there some theoretical result possible on the tolerance to outliers or Gaussian noise levels?
>
> Answer. Since our target application was for molecular graphs, the bounded noise is justified by small atomic vibrations, so atoms are real physical objects that cannot disappear (as outliers) or appear from "thin air". However, Problem 1.1 makes sense for any other data objects (instead of graphs) and equivalences (instead of isometry or rigid motion). If we consider graphs with outliers (vertices or edges), the first step is to define a suitable equivalence relation. After that Problem 1.1 will make sense.
>
> We are happy to answer any other questions.